# The CLIP Model is Secretly an Image-to-Prompt Converter

**Yuxuan Ding**[*]
School of Electronic Engineering
Xidian University
Xi'an 710071, China
yxding@stu.xidian.edu.cn

**Chunna Tian** [†]
School of Electronic Engineering
Xidian University
Xi'an 710071, China
chnatian@xidian.edu.cn

**Haoxuan Ding**
Unmanned System Research Institute
Northwestern Polytechnical University
Xi'an 710072, China
haoxuan.ding@mail.nwpu.edu.cn

**Lingqiao Liu** [†]
Australian Institute for Machine Learning
The University of Adelaide
Adelaide 5005, Australia
lingqiao.liu@adelaide.edu.au

## Abstract

The Stable Diffusion model is a prominent text-to-image generation model that relies on a text prompt as its input, which is encoded using the Contrastive Language-Image Pre-Training (CLIP). However, text prompts have limitations when it comes to incorporating implicit information from reference images. Existing methods have attempted to address this limitation by employing expensive training procedures involving millions of training samples for image-to-image generation. In contrast, this paper demonstrates that the CLIP model, as utilized in Stable Diffusion, inherently possesses the ability to instantaneously convert images into text prompts. Such an image-to-prompt conversion can be achieved by utilizing a linear projection matrix that is calculated in a closed form. Moreover, the paper showcases that this capability can be further enhanced by either utilizing a small amount of similar-domain training data (approximately 100 images) or incorporating several online training steps (around 30 iterations) on the reference images. By leveraging these approaches, the proposed method offers a simple and flexible solution to bridge the gap between images and text prompts. This methodology can be applied to various tasks such as image variation and image editing, facilitating more effective and seamless interaction between images and textual prompts.

## 1   Introduction

In recent years, there has been a surge of interest in vision-and-language research, particularly in the field of text-to-image generation. Prominent models in this domain include autoregression models like DALL-E [1] and Make-A-Scene [2], as well as diffusion models like DALL-E 2 [3] and Stable Diffusion [4]. These models have revolutionized the quality of generated images. They leverage text prompts to synthesize images depicting various objects and scenes that align with the given text. Among these models, Stable Diffusion [4] stands out as a significant open-source model. It serves as a foundation for many recent works, including image generation [5, 6, 7, 8], image editing [9, 10, 11, 12, 13, 14], and more.

---

[*]This work was done while Yuxuan Ding was visiting The University of Adelaide as a visiting researcher.
[†]Corresponding author.

37th Conference on Neural Information Processing Systems (NeurIPS 2023).

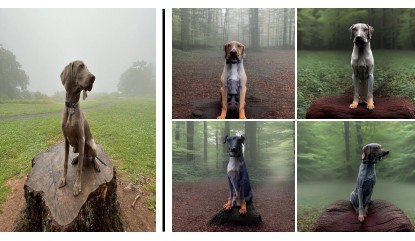

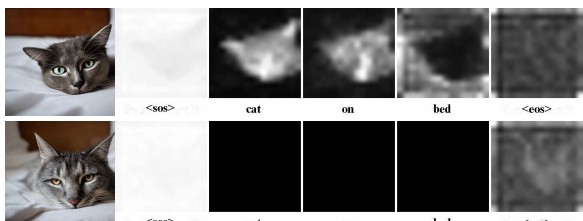

Figure 1: Demonstration of image variation. The image on the left is a real reference image, while the four on the right are generated from our method.

Figure 2: Attention map of Stable Diffusion [4]. The bottom row sets attention weights of caption words to zero, only keeping the start-/end-token, so the caption maps of the bottom are black. Also notice the start-token has strong weights so the map is all white.

However, text prompts have limitations when it comes to incorporating unspeakable information from reference images. It becomes challenging to generate a perfect and detailed prompt when users want to synthesize images related to a picture they have seen. Image variation techniques aim to address this limitation by enabling users to generate multiple variations of an input image, without relying on complex prompts. As illustrated in Fig. 1, the generated variations closely resemble the reference image, often sharing the same scene or objects but with distinct details.

Stable Diffusion Reimagine (SD-R) [4][3] is a recently proposed image variation algorithm. It achieves this goal by retraining Stable Diffusion [4], where the text encoder is replaced with an image encoder to adapt the model for image input. The model is trained using millions of images and over 200,000 GPU-hours, enabling it to effectively generate image variations based on reference images.

In this paper, we make a significant discovery that allows a more cost-effective image-to-prompt conversion approach. We find the CLIP model [15], as utilized in Stable Diffusion, can be repurposed as an effective image-to-prompt converter. This converter can be directly employed or served as a valuable initialization for a data-efficient fine-tuning process. As a result, the expenses associated with constructing or customizing an image-to-prompt converter can be substantially reduced.

More specifically, our method is built upon a surprising discovery: the control of image generation through text is primarily influenced by the embedding of the end-of-sentence (EOS) token. We found that masking all word tokens, except for the start and end tokens, does not adversely affect the quality of image generation, as illustrated in Figure 2. Simultaneously, during CLIP training, the projection of the end-token embedding is trained to align with the visual embedding. This inherent relationship enables us to derive a closed-form projection matrix that converts visual embedding into an embedding that is capable of controlling the generation of Stable Diffusion [4]. We call this method Stable Diffusion Image-to-Prompt Conversion (SD-IPC).

In addition, we introduce two methods to enhance the quality and flexibility of image-to-prompt conversion. The first approach involves parameter-efficient tuning using a small amount of data, consisting of only 100 images and requiring just 1 GPU-hour. This method encourages the model to better preserve image information and enables practitioners to control the specific content they want to retain when generating new images. The second approach involves customizing the model on reference images using a few iterations, ensuring that the generated images are closer to specific concepts. While this approach has been explored in previous research, we demonstrate that with the advantageous initialization provided by SD-IPC, the online fine-tuning requires significantly fewer iterations to achieve desirable results.

## 2 Background and Related Works

### 2.1 Diffusion Model

Firstly, we present a brief overview of the Stable Diffusion [4], which serves as our underlying model. Diffusion models (DMs) [16, 17, 18, 19] belong to a class of latent variable models. In DMs, there exist two Markov chains known as the *diffusion process* and the *reverse process*, both having a fixed

---

[3]https://stability.ai/blog/stable-diffusion-reimagine

length $T$. The diffusion process progressively introduces Gaussian noise to the original data $(\mathbf{x}_0)$ until the signal becomes corrupted $(\mathbf{x}_T)$. During DMs training, the reverse process is learned, which operates in the opposite direction of the diffusion process. The reverse process can be viewed as a denoising procedure, moving from $\mathbf{x}_t$ to $\mathbf{x}_{t-1}$ at each step. After multiple denoising steps, the model obtains instances that closely resemble the real data.

Stable Diffusion [4] is built on the Latent Diffusion Model (LDM) [4]. LDM [4] proposed to do diffusion process in a latent space rather than the usual pixel space, significantly reducing the training and inference cost of the diffusion model. The authors proposed to utilize a VAE compression to get the latent code $\mathbf{z}_0$, which is $\mathbf{x}_0$ above. Diffusion process will build on the latents. A U-Net architecture [17] with timestep and text conditions would do the reverse. The text prompt is injected into the model with cross-attention layers. We denote $\epsilon_\theta\left(\mathbf{z}_t, c_{txt}(p_{txt}), t\right)$ as the output of the U-Net, which is the predicted denoising result. $p_{txt}$ is the textual prompt and $c_{txt}(p_{txt})$ is the prompt embedding from the text encoder. $t$ is the timestep. The training objective of DMs is as followed:

$$\mathbb{E}_{\epsilon,\mathbf{z},p_{txt},t}\left[\left\|\epsilon - \epsilon_\theta\left(\mathbf{z}_t, c_{txt}(p_{txt}), t\right)\right\|_2^2\right], \tag{1}$$

where $\epsilon \sim \mathcal{N}\left(\mathbf{0}, \mathbf{I}\right)$ is the noise used to corrupt clean latent variables. During the generation, the latent $\mathbf{z}_t$, which starts at a random Gaussian noise $\mathbf{z}_T$, will recursively go through a denoising operation until $\mathbf{z}_0$ is sampled. Finally, $\mathbf{z}_0$ is reconstructed to an image by the VAE.

## 2.2 CLIP Model

The CLIP model [15] has garnered significant acclaim as a groundbreaking zero-shot model in recent years. Its training process demands optimizing a contrastive loss function using extensive 400-million pairs of images and corresponding text descriptions. Through the meticulous training, the model has been able to achieve unparalleled capabilities in zero-shot classification and image-text retrieval.

The model comprises an image encoder $\text{CLIP}_i\left(\cdot\right)$, a text encoder $\text{CLIP}_t\left(\cdot\right)$, a visual projection layer $W_i$, and a textual projection layer $W_t$. The image encoder encodes an input image $x$ into a visual embedding $\mathbf{f}_{img}$ derived from a special class-token. By applying the visual projection layer, the embedding is projected into the CLIP visual embedding $\mathbf{f}_{img}^c$. Similarly, the text encoder processes the input text, yielding a sequence of output embeddings $\mathbf{f}_{txt}$ for each text token and a start token and end-of-sentence (EOS) )token. The embedding of the EOS token $\mathbf{f}_{txt}^{t,\langle eos\rangle}$, where $t$ denotes the length of the sentence, is projected into the CLIP textual embedding $\mathbf{f}_{txt}^c$ through $W_t$. Formally,

$$\mathbf{f}_{img} = \text{CLIP}_i\left(x\right), \quad \mathbf{f}_{img}^c = W_i \cdot \mathbf{f}_{img}, \tag{2}$$

$$\mathbf{f}_{txt} = \text{CLIP}_t\left(s\right), \quad \mathbf{f}_{txt}^c = W_t \cdot \mathbf{f}_{txt}^{t,\langle eos\rangle}. \tag{3}$$

The training objective of CLIP is to maximize the cosine similarity between $\mathbf{f}_{txt}^c$ and $\mathbf{f}_{img}^c$ for matched sentence-image pair while minimizing this similarity for unmatched pairs. For the simplicity of discussion, we denote the space spanned by $\mathbf{f}_{txt}$ as $\mathcal{T}$-space and the space spanned by $\mathbf{f}_*^c$ as $\mathcal{C}$-space.

The CLIP text encoder [15] is directly used in Stable Diffusion to encode text prompts. It encodes a text prompt as a sequence of embeddings:

$$\mathbf{f}_{txt} := \left[\mathbf{f}_{txt}^{0,\langle sos\rangle}, \mathbf{f}_{txt}^{1,w_0}, ..., \mathbf{f}_{txt}^{t,\langle eos\rangle}, ..., \mathbf{f}_{txt}^{76,\langle eos\rangle}\right] \tag{4}$$

where $\mathbf{f}_{txt}^{0,\langle sos\rangle}$, $\mathbf{f}_{txt}^{i,w}$ and $\mathbf{f}_{txt}^{t,\langle eos\rangle}$ denote the embeddings corresponding to the start-token, the $i$-th word token and end-token, respectively. From $\mathbf{f}_{txt}^{t+1,\langle eos\rangle}$ to $\mathbf{f}_{txt}^{76,\langle eos\rangle}$ are padded tokens.

## 2.3 Image Variation & Customized Generation

**Image Variation.** Image variation aims to generate images similar to the reference image but not identical. SD-R [4] is proposed to address this problem, which builds upon the Stable-unCLIP model[4]. The authors fine-tuned the Stable Diffusion model [4] to align with the CLIP visual embedding. In SD-R [4], images can be directly input into the diffusion model through CLIP image encoder. Since the original Stable Diffusion is conditioned on text only, an expensive fine-tuning is required to

---

[4] https://huggingface.co/stabilityai/stable-diffusion-2-1-unclip

accommodate this new input. The process took 200,000 GPU-hours on NVIDIA A100-40GB GPU while our approach only requires 1 GPU-hour on NVIDIA A5000-24GB GPU[5].

**Customized Generation.** Recent works such as DreamBooth [11], Textual Inversion [14], and Custom Diffusion [13] focus on learning a special text prompt to feature specific objects or persons from the reference images. For instance, given several photos of a particular cat, these methods use a special-token "$\langle s \rangle$ cat" to represent the concept and incorporate it with the text prompt. DreamBooth [11] and Custom Diffusion [13] also perform simultaneous fine-tuning of diffusion model parameters. However, the fine-tuning process is still somewhat time-consuming, with Custom Diffusion [13] requiring nearly 6 minutes on 2 NVIDIA A100 GPUs. In contrast, our fast update SD-IPC only needs 1 minute on 2 A5000 GPUs.

**Image Editing.** Stable Diffusion [4] is commonly used for image editing tasks. Prompt-to-Prompt [9] and Plug-and-Play [10] utilize attention map as a bridge to enable concept and style manipulation. Null-Text Inversion [20] and Pix2Pix-Zero [21] relies on inversion-based methods. InstructPix2Pix [22] creates a dataset of paired edited images and fine-tunes Stable Diffusion [4] as an editing model. It's important to highlight that while our primary focus in developing this method was to enhance image variation, it can also be employed to generate images based on prompts that combine both textual instructions and accompanying images. Notably, unlike existing approaches that frequently reproduce the layout of the original image in the generated output, our method operates without being confined to replicating the exact layout of the source image.

## 3 Methodology

### 3.1 Image-to-Prompt Conversion via Projecting CLIP embedding

By design, the image generation process in the stable diffusion model should be influenced by embeddings of all tokens in a prompt, like Eq. (4). Interestingly, we have discovered that masking word tokens, by setting their attention weights to 0 except for the start-/end-token, does not have a negative impact on the quality of generated images. This finding is visually illustrated in Figure 2.

On another note, the training objective of CLIP [15] is to match the embeddings $\mathbf{f}_{img}^c$ and $\mathbf{f}_{txt}^c$, with $\mathbf{f}_{txt}^c$ being essentially a projection of $\mathbf{f}_{txt}^{t,\langle eos \rangle}$. This inherent relationship, coupled with the aforementioned observation, leads us to establish a connection between $\mathbf{f}_{img}^c$ and $\mathbf{f}_{txt}^{t,\langle eos \rangle}$, effectively converting the visual embedding to a prompt embedding.

Formally, we assume that after training, CLIP model can induce high cosine similarity between the $\mathbf{f}_{img}^c$ and $\mathbf{f}_{txt}$ and we can further make the following approximation:

$$\frac{\mathbf{f}_{img}^c}{\|\mathbf{f}_{img}^c\|} \approx \frac{\mathbf{f}_{txt}^c}{\|\mathbf{f}_{txt}^c\|}, \quad \text{with } \mathbf{f}_{txt}^c = W_t \mathbf{f}_{txt}^{t,\langle eos \rangle}. \tag{5}$$

By using Moore-Penrose pseudo-inverse [23] on $W_t$[6], we obtain an estimate of $\mathbf{f}_{txt}^{t,\langle eos \rangle}$ from $\mathbf{f}_{img}^c$:

$$\mathbf{f}_{txt}^{t,\langle eos \rangle} \approx \frac{\|\mathbf{f}_{txt}^c\|}{\|\mathbf{f}_{img}^c\|} W_t^+ \mathbf{f}_{img}^c := \mathbf{f}_{txt}^{cnvrt}, \quad \text{where, } W_t^+ = \left(W_t^\top W_t\right)^{-1} W_t^\top, \tag{6}$$

where we empirically observe $\|\mathbf{f}_{txt}^c\|$ can be well approximated by a constant, e.g., $\|\mathbf{f}_{txt}^c\| = 27$ and $W_t$ can be obtained from the pretrained CLIP model [15]. We denote the converted embedding as $\mathbf{f}_{txt}^{cnvrt}$ and use it to assemble a pseudo-prompt sequence with the following format:

$$\tilde{\mathbf{f}}_{txt} := \left[\mathbf{f}_{txt}^{0,\langle sos \rangle}, \mathbf{f}_{txt}^{1,cnvrt}, ..., \mathbf{f}_{txt}^{76,cnvrt}\right], \tag{7}$$

where $\mathbf{f}_{txt}^{1,cnvrt} = \cdots = \mathbf{f}_{txt}^{76,cnvrt} = \mathbf{f}_{txt}^{cnvrt}$. In other words, we replace all word-tokens, pad-tokens and end-token in Eq. (4) with the converted $\mathbf{f}_{txt}^{cnvrt}$, based on the fact that $\mathbf{f}_{txt}^{cnvrt}$ is an approximation of $\mathbf{f}_{txt}^{t,\langle eos \rangle}$ and masking word-tokens does not influence the generation[7].

---

[5]FP16 computing performance of A100 is 77.97 TFLOPS vurse 27.77 TFLOPS of A5000.

[6]We use singular value decomposition (SVD) to form the pseudo-inverse, singular values which are smaller than 0.3 will be treated as 0.

[7]Here we keep the pad-tokens, so the index of token is from 0 to 76, where the maximum length of CLIP text input is 77, even they are the same $\mathbf{f}_{txt}^{cnvrt}$, they would contribute to the attention weights in cross-attention, to decrease the weights of start-token.

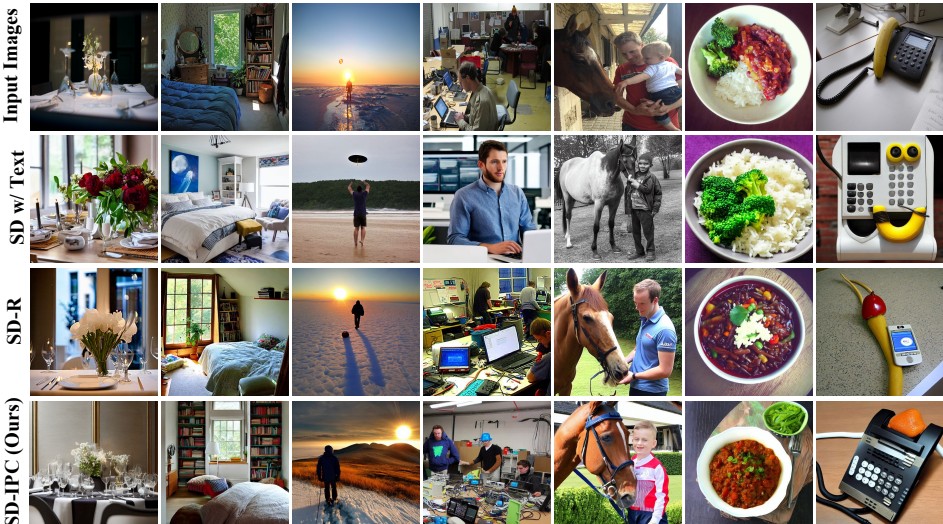

Figure 3: Image variation results on MSCOCO [24]. SD w/ Text [4] is generation from the ground-truth text prompts that are not available for variation methods such as SD-R and SD-IPC. SD-IPC is our method, notice that SD-IPC does not need any training compared to SD-R [4].

This approximation allows immediate conversion of an image to a text prompt by directly mapping it to an (approximately) equivalent prompt. **We refer to this method as Stable Diffusion Image-to-Prompt Conversion (SD-IPC).** Experimental results in Fig. 3 demonstrate that SD-IPC effectively captures the semantic information present in the reference image and enables image variation.

Furthermore, we have identified a simple yet effective approach to combine both the text prompt and the converted image prompt within our framework. To achieve this, we perform a weighted average of the two embeddings. Formally, the process can be described as follows:

$$\mathbf{f}_{txt}^{comb} = \mathbf{f}_{txt}^{cnvrt} + \alpha \cdot \mathbf{f}_{txt}^{t,\langle eos \rangle}, \quad \tilde{\mathbf{f}}_{txt}^{edit} = \left[ \mathbf{f}_{txt}^{0,\langle sos \rangle}, \mathbf{f}_{txt}^{1,w_0}, ..., \mathbf{f}_{txt}^{t,comb}, ..., \mathbf{f}_{txt}^{76,comb} \right], \tag{8}$$

where $\mathbf{f}_{text}^{i,comb} = \mathbf{f}_{text}^{comb}$ is the combined-token embedding and $\alpha$ is a hyperparameter to control the expression of editing text. Notice that the editing word-token $\mathbf{f}_{txt}^{i,w}$ are also in the embedding sequence. Conditioning on $\tilde{\mathbf{f}}_{text}^{edit}$ could generate images that match both the visual and textual conditions. We report some editing results in Appendix D.2.

## 3.2 Fine-tuning with Image-to-Prompt Conversion

While the aforementioned SD-IPC method demonstrates reasonable performance, it still faces challenges when it comes to real-world applications due to two main reasons. Firstly, the conversion process in SD-IPC relies on approximations, which may not always yield optimal results. Secondly, determining the exact topic or theme of an image introduces ambiguity. As the saying goes, "an image is worth a thousand words", but precisely which words? The same reference image can be interpreted differently based on its objects, scenes, styles, or the identities of people depicted within. Therefore, it becomes crucial to have a method that allows control of the content we wish to preserve and convert into the prompt. To address these concerns and cater to the needs, we propose a partial fine-tuning approach for the CLIP converter derived from Sec. 3.1.

In proposed approach, we focus on fine-tuning two specific types of parameters. Firstly, we address the optimization of the projection matrix within the cross-attention layer of the U-Net in Stable Diffusion [4]. This aspect aligns with the methodology employed in Custom Diffusion [13]. Furthermore, we incorporate deep prompt tuning [25] into the transformer of the CLIP image encoder. Deep prompt tuning [25] introduces learnable tokens within all layers of the transformer while keeping the weights of other components fixed. More details can be found in Appendix A.

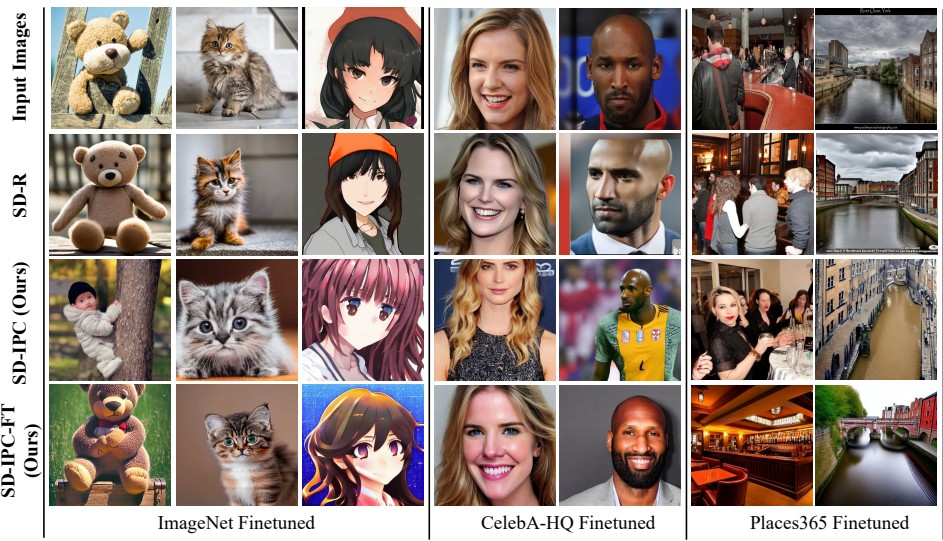

Figure 4: Fine-tuned SD-IPC, denoted as SD-IPC-FT, can enhance the image-to-prompt conversion quality.

The parameters can be learned by using the following loss:

$$\mathbb{E}_{\epsilon,\mathbf{z},x_{\text{ref}},t}\left[\|\epsilon - \epsilon_\theta\left(\mathbf{z}_t, c_{img}(x_{\text{ref}}),t\right)\|^2\right] + \mathbb{E}_{\epsilon,\mathbf{z},p_{txt},t}\left[\|\epsilon - \epsilon_\theta\left(\mathbf{z}_t, c_{txt}(p_{txt}),t\right)\|^2\right], \qquad (9)$$

here the first term is $\mathcal{L}_{cnvrt}$, which is fine-tuning with the image-to-prompt input, and the second term is $\mathcal{L}_{text}$, which is the original text-to-image training loss, we utilize this as a regularization to keep the text-to-image generation. In the proposed approach, $c_{img}(\cdot)$ refers to the image-to-prompt conversion derived from SD-IPC. It encompasses the CLIP image transformer augmented with the newly-introduced learnable prompts in deep prompting and the fixed inverse matrix derived from Eq. (6). During tuning, the inverse projection matrix remains unchanged. $\mathbf{z}_t$ represents the latent representation of the target image $x_{\text{target}}$ at time step $t$. The objective function aims to encourage the image-to-prompt conversion to extract information from $x_{\text{ref}}$ that facilitates the recovery of $x_{\text{target}}$. There are two possible choices for $x_{\text{target}}$: (1) $x_{\text{target}}$ can be selected to be the same as $x_{\text{ref}}$. (2) $x_{\text{target}}$ and $x_{\text{ref}}$ can be different images, but with a shared visual concept that we intend to extract as the prompt. This usually poses stronger supervision to encourage the converter to extract information related to the shared theme. The schematic representation of this scheme is illustrated in Appendix C.2.

We use images randomly sampled from ImageNet [26], CelebA-HQ [27], and Places365 [28] dataset to encourage the model extract object, identity, and scene information, respectively. Experiments show that merely 100 images and 1 GPU-hour of training are sufficient for achieving satisfied results thanks to the good initialization provided by SD-IPC. **We call this approach SD-IPC-FT**, the results are shown in Fig. 4. Some editing examples are listed in Fig. 5, Fig. 6, and Appendix D.4.

### 3.3 Fast Update for Customized Generation

Existing methods, such as DreamBooth [11] and Custom Diffusion [13], suggest that partially fine-tuning the model on given concept images before generation can be an effective way to synthesized images with customized visual concepts, *e.g.*, people with the same identity. Our approach can also benefit from this scheme by performing such an online update with SD-IPC. This can be achieved by simply replacing the training images in SD-IPC-FT with reference images and use $\mathcal{L}_{convrt}$ only. **We call this method SD-IPC-CT** (CT stands for customized concept tuning). Interestingly, we find that our method can generate customized images with much fewer updates. As a comparison, SD-IPC-CT only takes 30-iteration updates with around 1 minute on 2 A5000 GPUs while the Custom Diffusion [13] needs 250 iterations (6 minutes on 2 A100 GPUs). We report customized generation in Fig. 7.

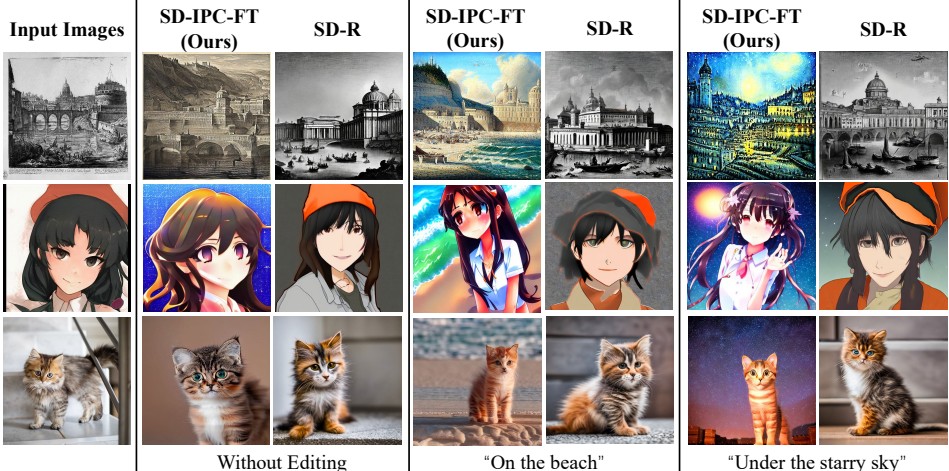

| Input Images | SD-IPC-FT (Ours) | SD-R | SD-IPC-FT (Ours) | SD-R | SD-IPC-FT (Ours) | SD-R |
|---|---|---|---|---|---|---|
| | Without Editing | | "On the beach" | | "Under the starry sky" | |

Figure 5: Image editing result with SD-IPC-FT trained with 100 images sampled from ImageNet [26]. SD-IPC-FT shows better editing performance than that of SD-R [4].

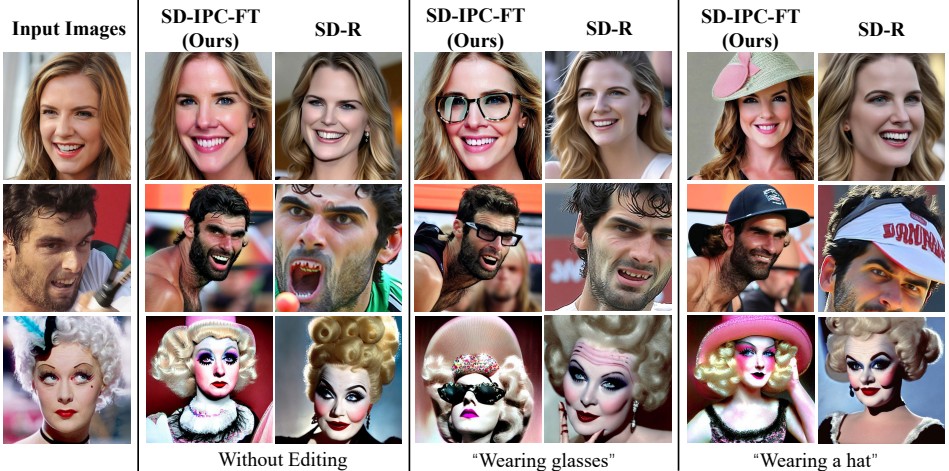

| Input Images | SD-IPC-FT (Ours) | SD-R | SD-IPC-FT (Ours) | SD-R | SD-IPC-FT (Ours) | SD-R |
|---|---|---|---|---|---|---|
| | Without Editing | | "Wearing glasses" | | "Wearing a hat" | |

Figure 6: Image editing result with SD-IPC-FT trained with 100 images sampled from CelebA-HQ [27]. SD-IPC-FT shows better editing performance than that of SD-R [4].

## 4 Experiments

### 4.1 Training Details

**Datasets & Evaluations.** In previous discussion, we propose three different fine-tuning schemes, using ImageNet [26] for object understanding, CelebA-HQ [27] for portrait understanding, and Places365 [28] for scene understanding. The specific training classes or identities we have selected for each dataset can be found in Appendix B. Each dataset includes 100 images, the test images are non-overlap with the training classes. In order to enable customized generation, we choose two objects and two identities as examples, each accompanied by five images. To assess the quality and semantic consistency of our generated outputs, we measure FID-Score [29] and CLIP-Score [30].

**Architecture & Hyperparameter.** We utilize Stable Diffusion v1.4[8] and CLIP ViT-L/14[9] models in this paper. We compare our method with the larger Stable-unCLIP-small model[10] using CLIP ViT-H/14 and a 1,024-dimensional attention feature. Our method uses DDIM [19] for sampling, while

---

[8]https://huggingface.co/CompVis/stable-diffusion-v1-4

[9]https://huggingface.co/openai/clip-vit-large-patch14

[10]https://huggingface.co/stabilityai/stable-diffusion-2-1-unclip-small

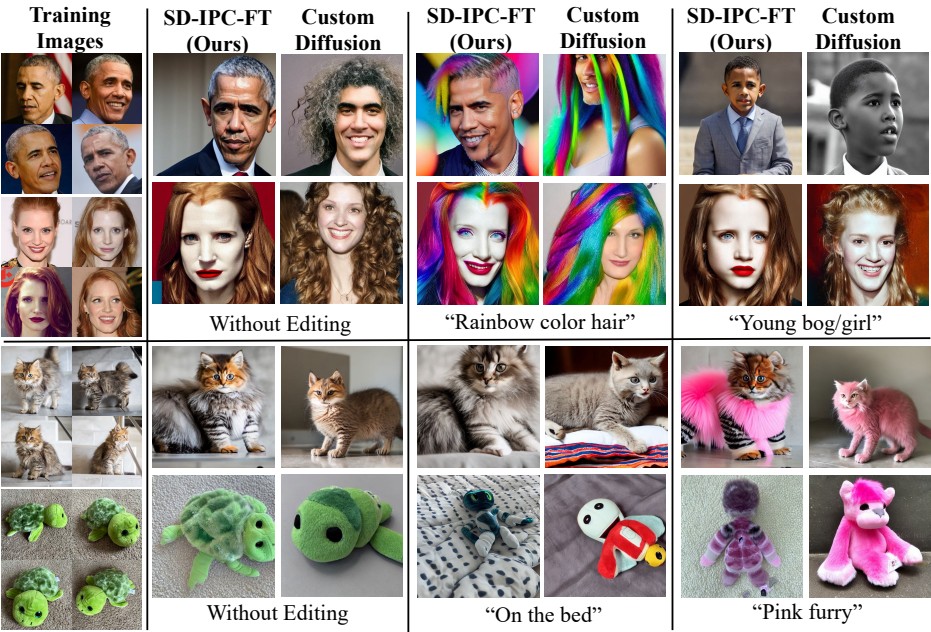

Figure 7: Customized generation examples. The images at left are training images, they are all from one concept or one identity. We compared our SD-IPC-CT with Custom Diffusion [13], notice that both results are trained by 5 reference images with merely 30 iterations.

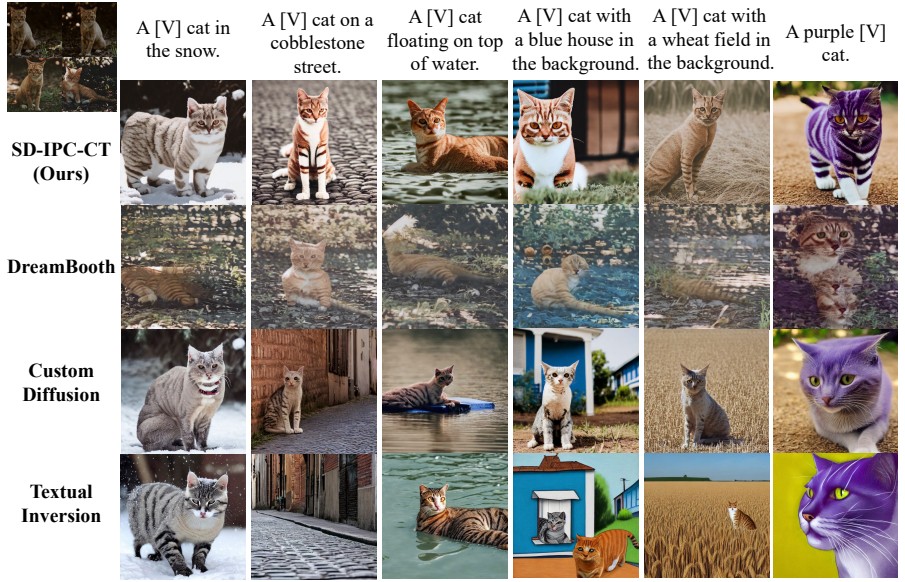

Figure 8: Results of DreamBooth [11] benchmark, the training images are listed at top-left corner.

Stable-unCLIP uses PNDM [31], both with 50 sampling steps. SD-IPC-FT is trained for 100, 50, and 100 epochs on ImageNet [26], CelebA-HQ [27], and Places365 [28], respectively. The learning rates for all datasets are $1e$-5 with cosine decay. Customized generation has a constant learning rate of $5e$-6 for 30-iteration updates. Training is conducted on 2 A5000 GPUs. The editing $\alpha$ is set to 0.9.

## 4.2 Image Variation Results

**SD-IPC.** We evaluate image variation on MSCOCO [24] using all 5,000 images in the 2017-split validation set. Fig. 3 compares text-based generation, SD-R [4], and our SD-IPC. Both SD-R [4]

Table 1: Accuracy and Retrieval recalls (%) of original CLIP [15] ($\mathcal{C}$-space) and the inverse matrix transfer ($\mathcal{T}$-space). Acc@$k$ is the top-$k$ accuracy of ImageNet [26] val-set, TR@$k$ and IR@$k$ are top-$k$ text and image retrieval recalls. This is a surprising result that there is almost no performance decline.

| Emb. Space | Acc@1 | Acc@5 | TR@1 | TR@5 | IR@1 | IR@5 |
|---|---|---|---|---|---|---|
| $\mathcal{C}$-space | 71.41 | 91.78 | 74.58 | 92.98 | 55.54 | 82.39 |
| $\mathcal{T}$-space | 69.48 | 90.62 | 71.62 | 92.06 | 54.82 | 82.20 |

Table 2: FID-Score [29] and CLIP-Score [30] (%) of the generation results. SD w/ Text [4] means the generation from ground-truth text.

| Methods | FID | CLIP-Score |
|---|---|---|
| SD w/ Text [4] | 23.65 | 70.15 |
| SD-R [4] | 19.86 | 82.59 |
| SD-IPC (Ours) | 24.78 | 73.57 |

and our method demonstrate image variation, but our approach utilizes an inverse matrix without training. The FID-Score [29] and CLIP-Score [30] are reported in Tab. 2. Our SD-IPC maintains image quality similar to text-based generation (FID-Score: 24.78 vs. 23.65, CLIP-Score: 73.57 vs. 70.15). Note that SD-R [4] achieves better results due to its advanced backbone model.

**Finetuning.** In Fig. 4, it is observed that SD-IPC may exhibit inconsistency, for example, the input "teddybear" generates a picture of "kid". This inconsistency could be attributed to fact that SD-IPC fails to discern the semantically related concepts "kid" and "teddybear". However, this issue can be rectified through fine-tuning, as demonstrated in Fig. 4, SD-IPC-FT achieves improved generation quality. Moreover, we find that the editing capability of SD-IPC-FT, as illustrated in Fig. 5, Fig. 6, and Appendix D.4, surpasses that of SD-R [4], and fine-tuning does not impact the editing performance. We incorporated a quantitative experiment to validate the superior editing performance of SD-IPC-FT in comparison to SD-R [4]. We utilize images from DreamBooth [11] benchmark and randomly select an editing text for each test image. Editing performance is evaluated using CLIP-T score, the quantitative results are presented in Tab. 4. As seen, our method achieves a higher CLIP-T score than SD-R [4]. Furthermore, we include the training-free SD-IPC for comparison, revealing even SD-IPC slightly outperforms SD-R [4]. The role of different target images used in the fine-tuning stage is outlined in Appendix C.2, which showcases how the choice of target images influences the generation result. Additional generation results are presented in Appendix D.

## 4.3 Customized Generation Results

In Fig. 7, we compare our SD-IPC-CT with Custom Diffusion [13] in terms of customized generation. We evaluate customization for two identities ("Obama" and "Chastain") and two objects ("cat" and "tortoise"). The training process only requires 5 images and 30 iterations. The results in Fig. 7 reveal that Custom Diffusion [13] struggles to learn the details of the concept with such limited updates, while our SD-IPC-CT demonstrates impressive performance, particularly in the "young boy/girl" editing. However, for rare instances like the "tortoise" example, both methods do not perform well. For quantitative results, we followed DreamBooth [11]. We used DINO and CLIP-I for subject fidelity and CLIP-T for editing performance. Comparing with DreamBooth [11], Textual Inversion [14], and Custom Diffusion [13], the results are in Tab. 3, Fig. 8, and Appendix D.5. DreamBooth [11] excels in DINO/CLIP-I scores but lags in CLIP-T, indicating limited editing performance, evident in visually similar outputs to training images. Textual Inversion [14] and Custom Diffusion [13] have strong CLIP-T but weak DINO/CLIP-I scores, indicating challenges in preserving subject details. Our SD-IPC-CT method strikes a balance between subject identity preservation and editing performance.

Table 3: Results of DreamBooth [11] benchmark and the comparison with common methods.

| Method | DNIO | CLIP-I | CLIP-T | Comments |
|---|---|---|---|---|
| DreamBooth [11] | **60.11** | **77.78** | 25.81 | **Good Identity**, Weak Editing |
| Textual Inversion [14] | 25.11 | 62.44 | 29.53 | Weak Identity, **Good Editing** |
| Custom Diffusion [13] | 39.67 | 68.37 | **30.90** | Weak Identity, **Good Editing** |
| SD-IPC-CT (Ours) | 50.25 | 74.59 | 28.14 | **Good Identity**, **Good Editing** |

Table 4: Superior editing performance of SD-IPC-FT.

| Method | CLIP-T |
|---|---|
| SD-IPC | 26.84 |
| SD-IPC-FT | **28.69** |
| SD-R [4] | 26.01 |

## 4.4 Ablation Study

**Effectiveness of Inverse Projection Matrix.** To evaluate the effectiveness of the inverse projection matrix in SD-IPC, we introduce a fully-connected layer instead of the inverse matrix in the image-to-prompt conversion, referred to as SD-IPC-FC and SD-IPC-FC(I), where (I) means to initialize with

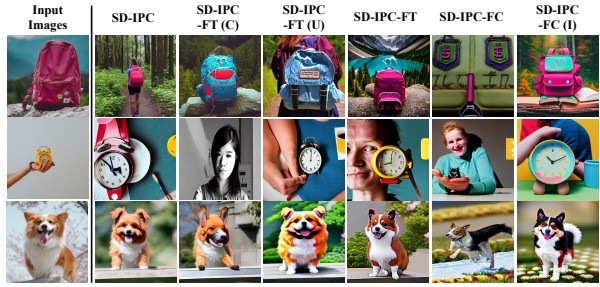

Figure 9: Image variation results of different fine-tuning settings. SD-IPC-FT (C) means only training CLIP prompts, SD-IPC-FT (U) means only training U-Net cross-attention layers, SD-IPC-FC (I) means initializing the FC-layer with the inverse matrix.

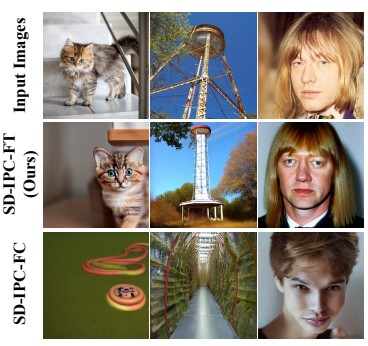

Figure 10: Effectiveness of using Eq. 6 for SD-IPC-FT.

our inverse projection matrix. We train the FC models with the same training data as SD-IPC-FT. However, the results in Fig. 10 indicate SD-IPC-FC suffers from overfitting. SD-IPC-FC(I) slightly alleviates the overfitting but still gets inferior results, shown in Fig. 9. This highlights that our SD-IPC-FT benefits from the good initialization of SD-IPC and preserves knowledge in CLIP [15].

**Prompt Learning & U-Net Fine-tuning.** We perform quantitative tests on (text-edited) image variation for the comprehensive ablation studies following the testing in Sec. 4.2. For text-edited variation, we use the editing text as the prompt, such as "A [Class Name] with a mountain in the background.". We present the results of individual fine-tuning for two components: SD-IPC-FT (C) for CLIP and SD-IPC-FT (U) for the U-Net. Qualitative results are available in Fig. 9, while quantitative results are provided in Tab. 5 and Tab. 6. It demonstrates that fine-tuning each component contributes to model adaptation, with the best performance achieved when simultaneously fine-tuning both two parts. Some editing comparisons are in Appendix C.3.

Additionally, we investigate the influence of the editing parameter $\alpha$ in Appendix C.1.



Table 5: Results of image variation with different fine-tuning settings.

| Method | DNIO | CLIP-I | CLIP-T |
|---|---|---|---|
| SD-IPC | 44.60 | 77.44 | 25.47 |
| SD-IPC-FT (C) | 49.11 | 76.51 | 25.82 |
| SD-IPC-FT (U) | 48.53 | 79.06 | **26.17** |
| SD-IPC-FT | **52.03** | **79.59** | 25.90 |

Table 6: Results of text-edited image variation with different fine-tuning settings.

| Method | DNIO | CLIP-I | CLIP-T |
|---|---|---|---|
| SD-IPC | 31.09 | 68.66 | 26.84 |
| SD-IPC-FT (C) | 29.10 | 67.03 | 27.99 |
| SD-IPC-FT (U) | 35.21 | 69.99 | 28.56 |
| SD-IPC-FT | **40.28** | **71.97** | **28.69** |



## 4.5 Limitations & Feature Directions

While SD-IPC offers an alternative to SD-R [4], there are remaining challenges. Firstly, the editing text must be contextually appropriate, as using "on the beach" to edit a portrait may result in a person being on the beach but lacking facial features. Secondly, SD-IPC currently does not support multiple image inputs. Another future study is to extend our method to generate a sequence of images with consistency. Appendix E shows some potential of our method in this direction.

## 5 Conclusion

This paper reveals that the CLIP model [15] serves as an image-to-prompt converter, enabling image variation in text-to-image Stable Diffusion [4] without extensive training. This finding enhances our understanding of the CLIP embedding space, demonstrating that a simple inverse matrix can convert visual embeddings into textual prompts. Leveraging this image-to-prompt conversion, our SD-IPC methods achieve impressive image variation and editing capabilities, while also enabling fast adaptation for customized generation. Experimental results also show the potential of our method in more multi-modal tasks. We anticipate that this study will inspire future research exploring the image-to-prompt pathway in CLIP-based or LDM-based models.

# 6 Acknowledgement

This work was partly supported by the China Scholarship Council under Grant 202006960047 and partly by the National Natural Science Foundation of China (No.62173265). Lingqiao Liu is supported by Centre of Augmented Reasoning.

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
