# A Demonstration of Architectures

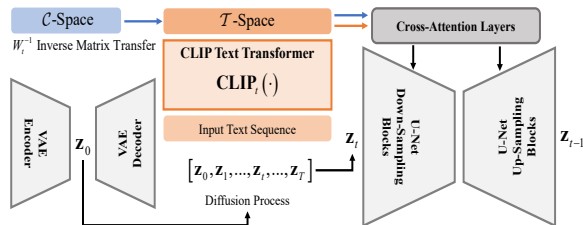 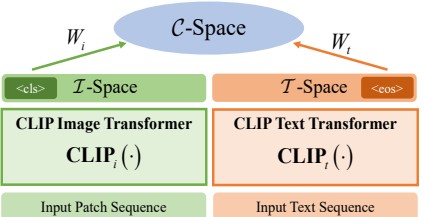

Figure 11: The architecture of Stable Diffusion [4]. Image is compressed by VAE to get the latent $\mathbf{z}_0$, then doing the diffusion process to acquire $\mathbf{z}_1 \sim \mathbf{z}_T$. The U-Net learns to predict removing noise $\epsilon_\theta(\mathbf{z}_t, \mathbf{c}, t)$ when input is $\mathbf{z}_t$. Notice that the text condition injects to the U-Net by cross-attention layers, and the blue dotted arrows present our reference image transfer.

Figure 12: The architecture of CLIP [15]. Class-token and end-token embeddings from $\mathcal{I}$-space and $\mathcal{T}$-space are projected into the $\mathcal{C}$-space, where the paired visual and textual embeddings are close. The Stable Diffusion [4] only utilizes the textual embedding from $\mathcal{T}$-space.

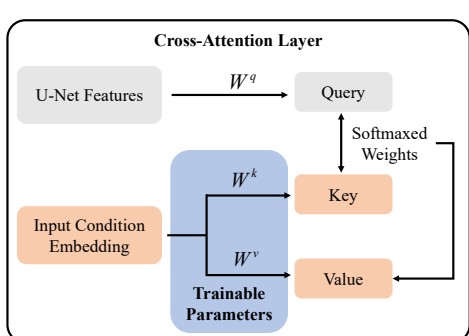 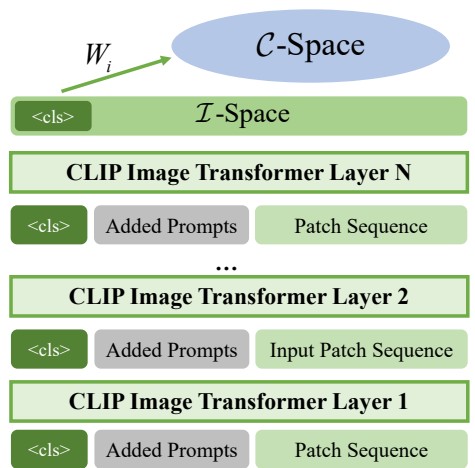

Figure 13: The cross-attention fine-tuning demonstration. $W^q, W^k, W^v$ are projections for query, key, and value, respectively. In Stable Diffusion [4], the query is U-Net feature, key and value are condition embedding (textual embedding or our converted embedding). We only fine-tune $W^k, W^v$ in updating, this is the same as [13].

Figure 14: Deep prompt tuning [25] for CLIP image transformer. The gray part is the added prompt in each layer. It will be optimized by our fine-tuning loss.

# B Fine-tuning Classes

Table 7: We randomly select 100 samples for each fine-tuning. This is the label list of the selected classes. For ImageNet [26] and Places365 [28], we select 20 classes with 5 images in each class. For CelebA-HQ [27], we select 10 people, below are their id-number in dataset.

| Dataset | fine-tuning Classes |
|---|---|
| ImageNet [26] | laptop computer, water jug, milk can, wardrobe, fox squirrel, shovel, joystick, wool, green mamba, llama, pizza, chambered nautilus, trifle, balance beam, paddle wheel |
| Places365 [28] | greenhouse, wet bar, clean room, golf course, rock arch, corridor, canyon, dining room, forest, shopping mall, baseball field, campus, beach house, art gallery, bus interior, gymnasium, glacier, nursing home, storage room, florist shop |
| CelebA-HQ [27] | 7423, 7319, 6632, 3338, 9178, 6461, 1725, 774, 5866, 7556 |

# C More Ablations

## C.1 Impact of Parameter $\alpha$

The impact of the editing parameter, $\alpha$, is examined in Fig. 15, focusing on the "Obama" customized model. A higher $\alpha$ value indicates a greater contribution of the editing text to the generation process. Different editing instructions may require different values of $\alpha$. For example, simpler edits like "wearing glasses" may be expressed with lower $\alpha$, even 0.0, as the added word-tokens in Eq. (8) also input the cross-attention.

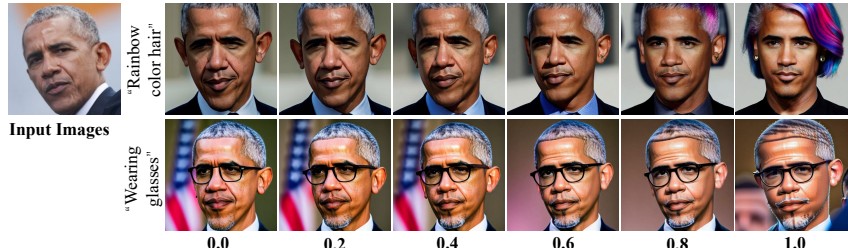

Figure 15: Editing with different $\alpha$ value. Higher $\alpha$ expresses more editing.

## C.2 Impact of different $x_{\text{target}}$

As introduced above, the reconstructed image $x_{\text{target}}$ can be $x_{\text{ref}}$ or another image. If we use an image from the same class but not $x_{\text{ref}}$ as $x_{\text{target}}$, we name this scheme A/B Training. Here are some comparisons of using A/B training or not. As shown in the first two columns of Fig. 16, ImageNet [26] is training data, if $x_{\text{target}}$ is $x_{\text{ref}}$ (w/o A/B training), the generated images would remain the "gray color" and "kid". If using A/B training, the images become "colorful castle" and emphasize the "laptop". Notably, the portrait results appear similar, as the $x_{\text{target}}$ resembles $x_{\text{ref}}$ even though it is another image in CelebA-HQ[27].

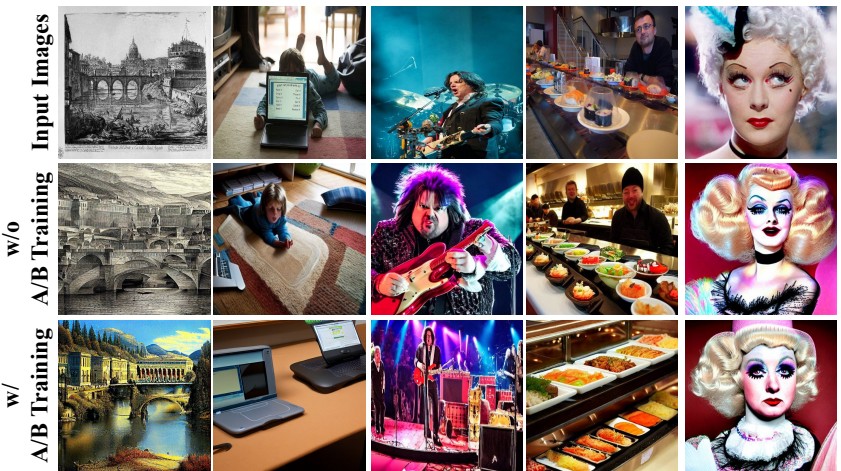

Figure 16: Demonstration of A/B training method.

## C.3 Editing Performance of Different Ablations

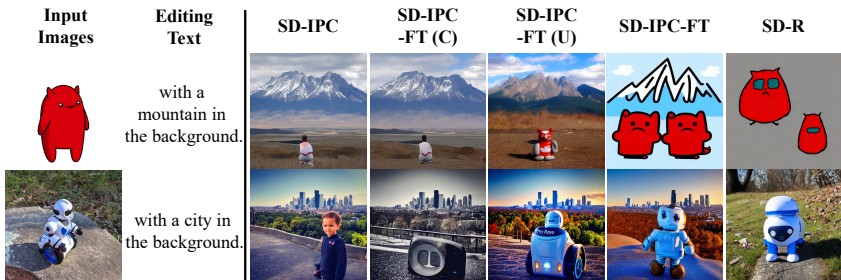

Figure 17: Results of text-edited image variation.

# D   More Results

## D.1   More MSCOCO [24] Variation of SD-IPC

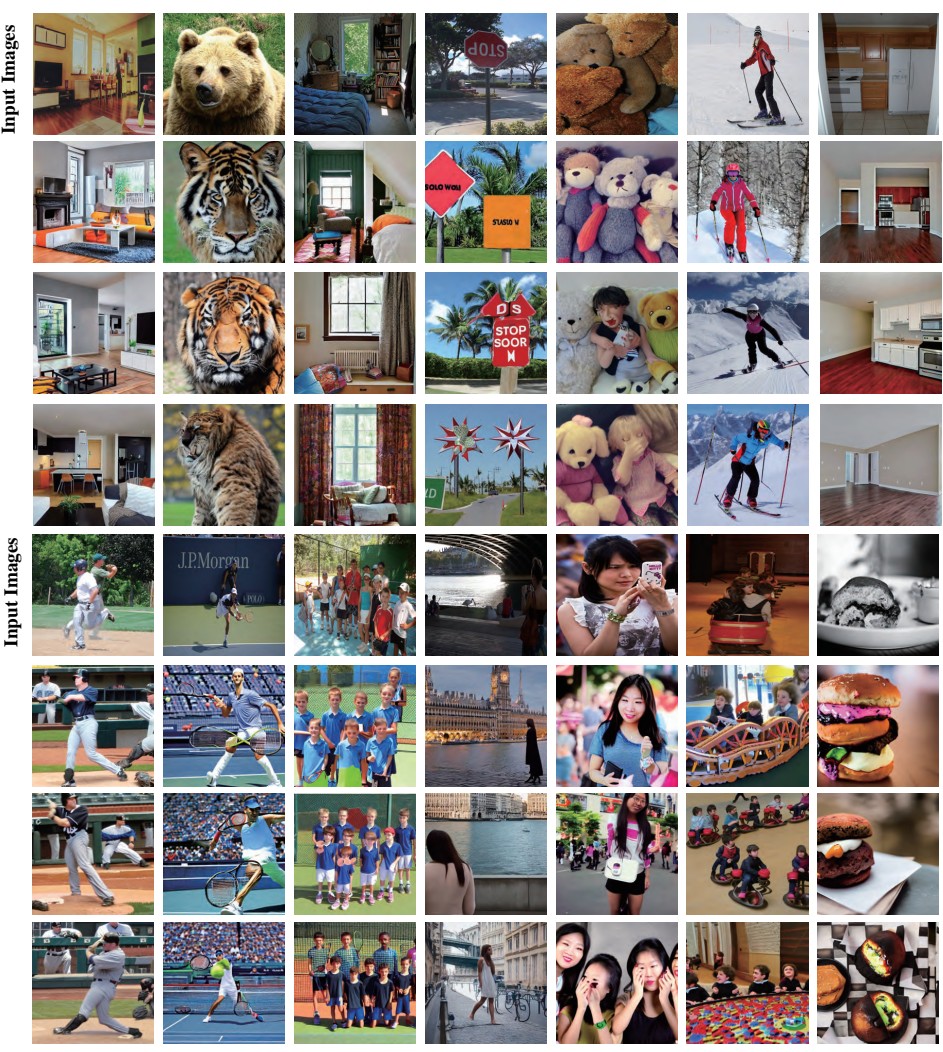

Figure 18: MSCOCO [24] variation with SD-IPC. We report three results for each input image.

## D.2 Editing with SD-IPC

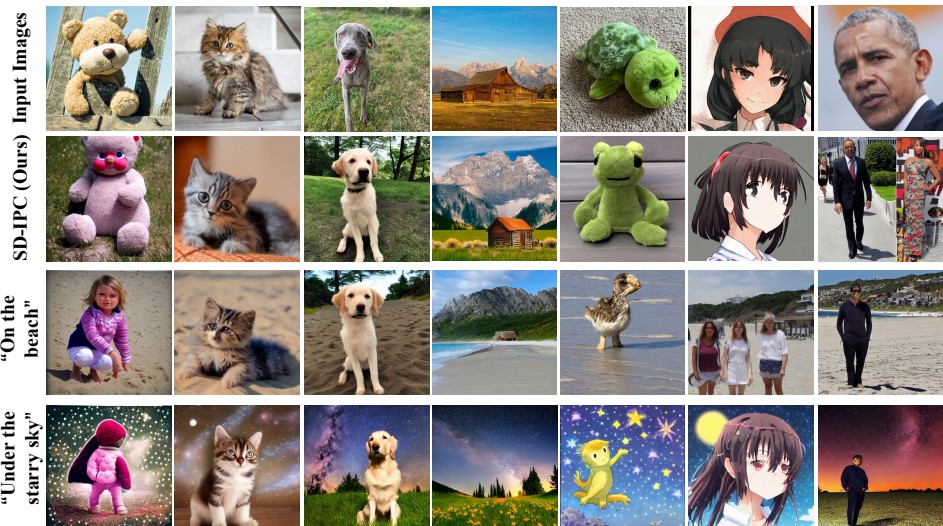

Figure 19: Object editing with SD-IPC.

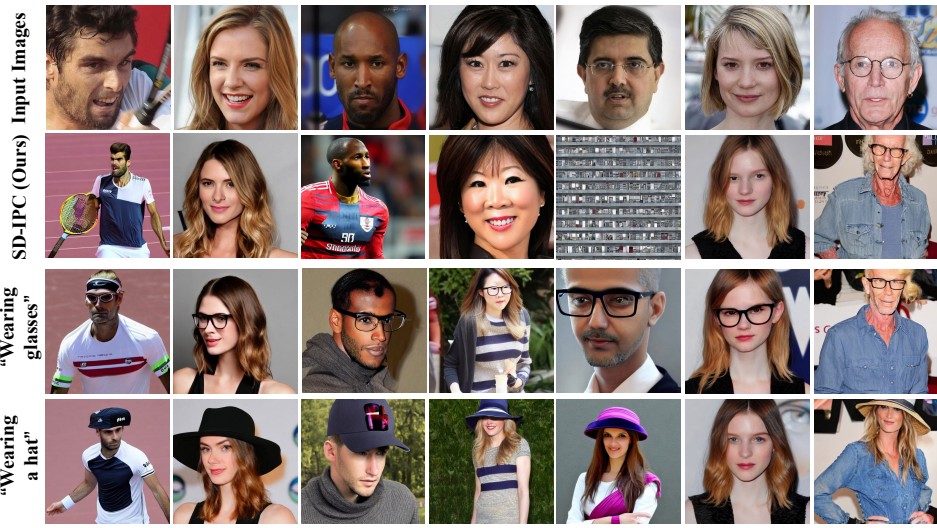

Figure 20: Portrait editing with SD-IPC.

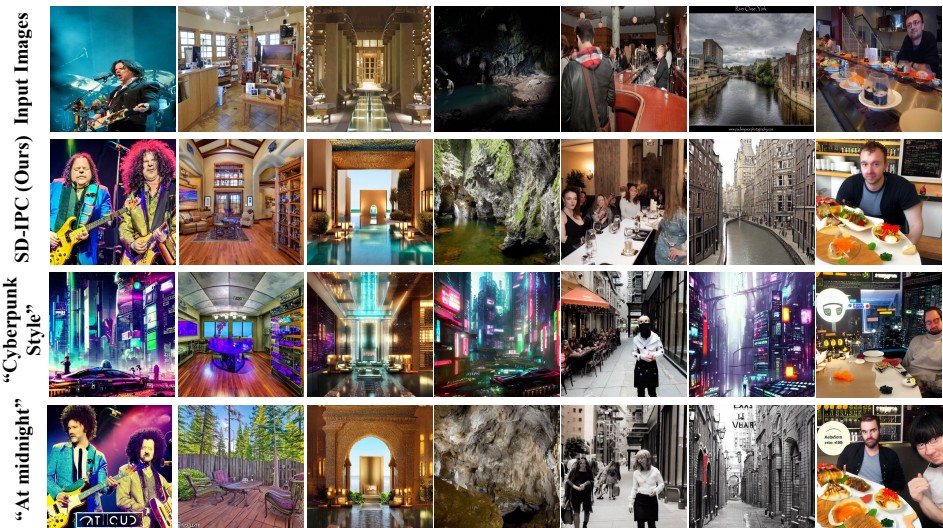

Figure 21: Scene editing with SD-IPC.

### D.3 More SD-IPC-FT Variation

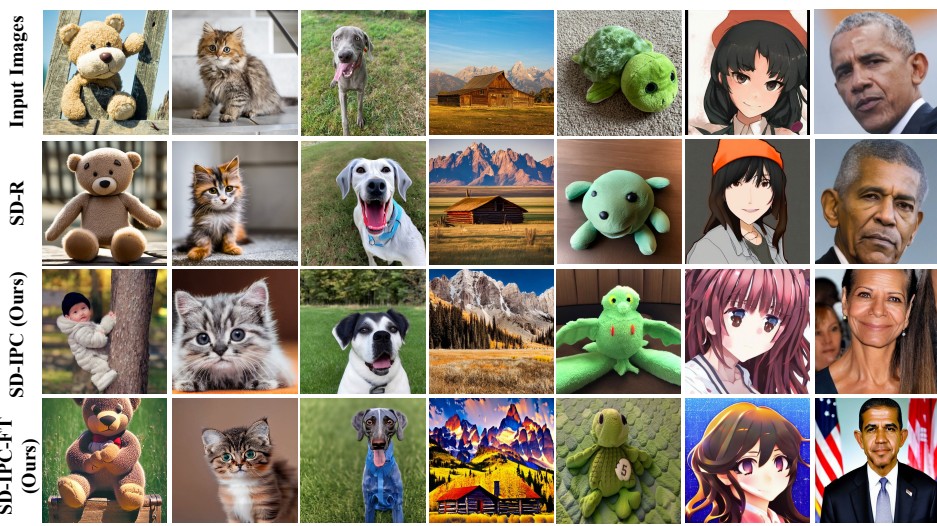

Figure 22: ImageNet [26] fine-tuned SD-IPC. There are some mistaken cases, such as the "teddybear" example above, in SD-IPC. But fine-tuning would fix the incompatible.

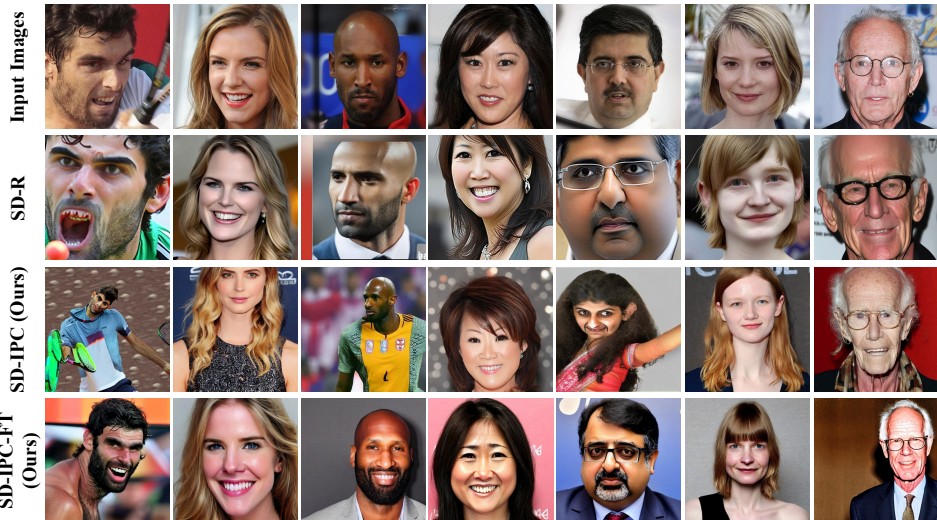

Figure 23: CelebA-HQ [27] fine-tuned SD-IPC. Despite that SD-IPC can generate portraits, its results coarsely match the semantics. SD-IPC-FT can create more similar portraits.

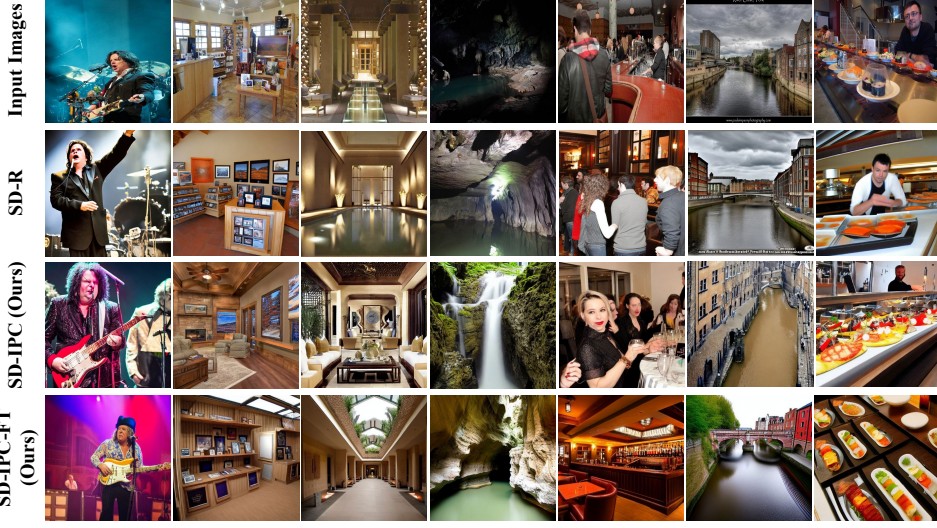

Figure 24: Places365 [28] fine-tuned SD-IPC. After fine-tuning, SD-IPC-FT can extract better scene features, such as the "cave" and "pub" examples.

## D.4 Editing Examples of Places365 Fine-tuned SD-IPC-FT

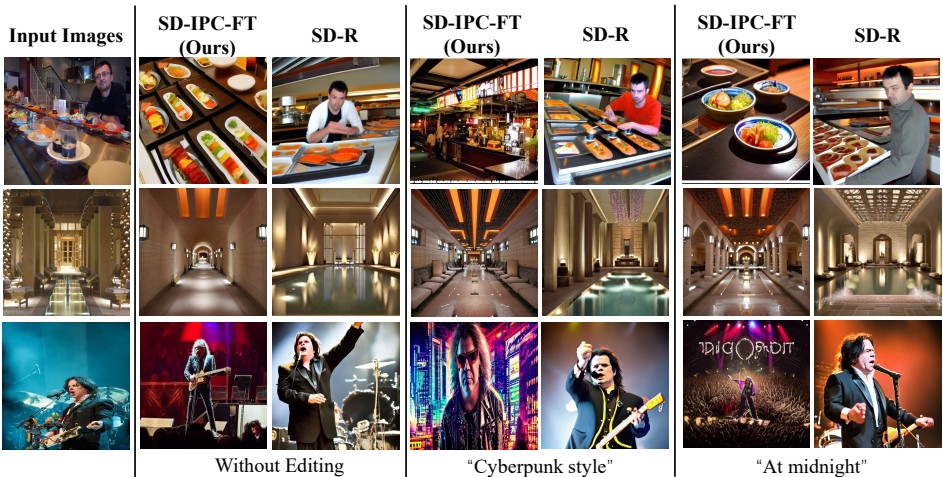

Figure 25: Places365 [28] fine-tuned SD-IPC-FT editing.

## D.5 More Comparisons of DreamBooth [11] Benchmark

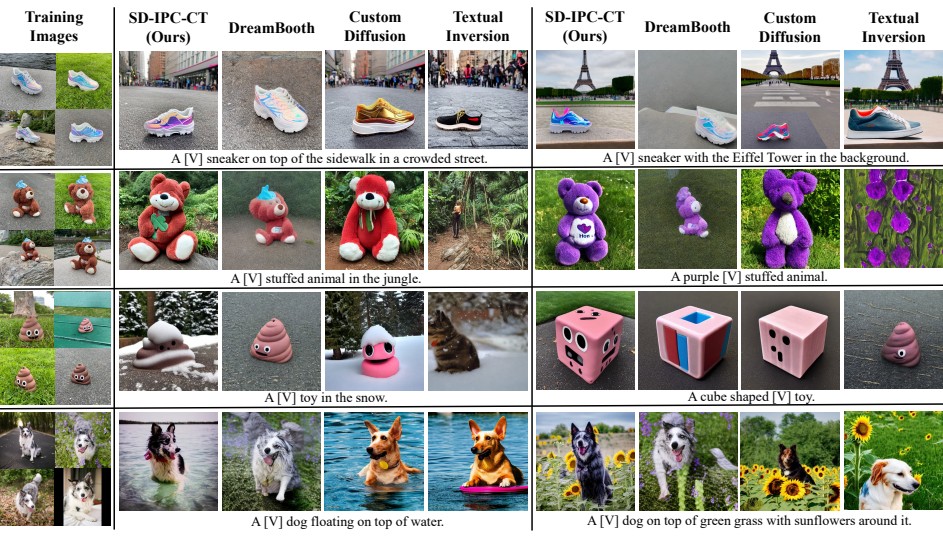

Figure 26: More results of DreamBooth [11] benchmark.

# E   Story Generation Example

- A little robot named Rusty went on an adventure to a big city.
- The robot found no other robot in the city but only people.
- The robot went to the village to find other robots.
- Then the robot went to the river.
- Finally, the robot found his friends.

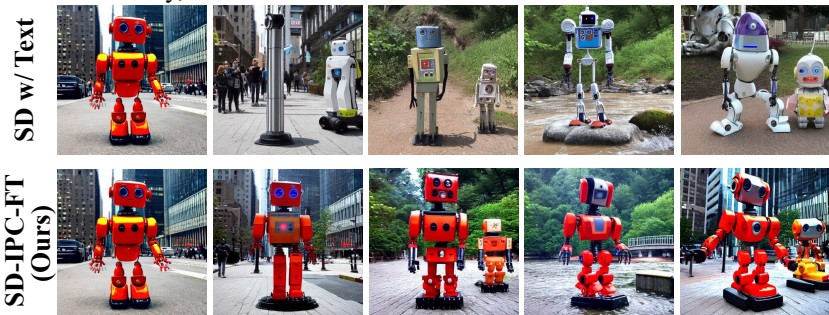

Figure 27: A story generation example with our ImageNet [26] fine-tuned SD-IPC-FT.

# F   Custom Diffusion [13] with More Updates

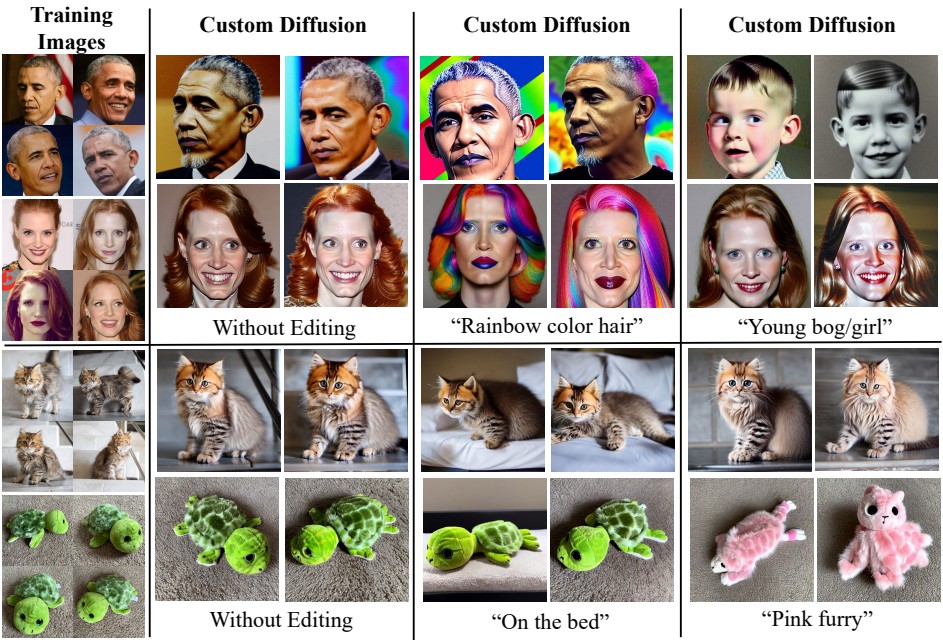

Figure 28: Following the training details in Custom Diffusion [13], we fine-tuned Custom Diffusion [13] with 250 iterations and shows it is effective for the generation and editing. We report 2 examples for each editing.