# OpenReview forum: "The CLIP Model is Secretly an Image-to-Prompt Converter"
_NeurIPS.cc/2023/Conference — NeurIPS 2023 poster_

### Official Review · Reviewer_hEWJ · 2023-07-06

**Soundness:** 3 good
**Presentation:** 2 fair
**Contribution:** 2 fair
**Rating:** 5
**Confidence:** 5

**Summary:**

This paper demonstrates that the CLIP model in Stable Diffusion inherently possesses the ability to convert images into text prompts, which can be achieved by utilizing a linear projection matrix that is calculated in a closed form.

**Strengths:**

1. The motivation of image-to-prompt conversion is clear and the proposed closed-form method is concise.
2. The capability of image-to-prompt conversion can be further enhanced by finetuning the model with a small amount of data and time.
3. The paper demonstrates various applications of the method.

**Weaknesses:**

1. The main concern lies in the experiments section:
(1) The quantitative comparison between SD-IPC and SD-IPC-FT can be given. And the quantitative and qualitative results when separately fine-tuning the two specific types of parameters (cross-attention layers or deep prompts) also can be given.
(2) Why not SD-IPC adopts the backbone of  SD-R, i.e.,  Stable Diffusion v2.1 ?
(3) In Sec.4.3, I suggest the paper adopts the dataset and edited prompt used by DreamBooth (https://github.com/google/dreambooth) to make a full comparison. Both qualitative and quantitative results should be presented. And the proposed method can also be compared with more methods (e.g., DreamBooth and Textual Inversion).
(4) It would be better if the quantitative results of the first ablation experiment are presented.
2. The motivation for applying deep prompts tuning in SD-IPC-FT needs to be further explained.
3. The editability of the proposed method may be limited.

**Questions:**

Please see the weaknesses. I am willing to improve my score if the concerns are addressed.

**Limitations:**

The limitations have been described in the paper.

---

> ### Author Rebuttal · Authors · 2023-08-10
>
> Thank you for the constructive comments. The common questions are first answered in the **General Responses**, then we clarify specific questions.
> ___
> ### Q1: The quantitative comparison between SD-IPC and SD-IPC-FT can be given. And the quantitative and qualitative results when separately fine-tuning the two specific types of parameters (cross-attention layers or deep prompts) also can be given.
> We appreciate your valuable comment. For the comprehensive ablation studies, we perform quantitative tests on image variation and text-edited image variation using images from the benchmark in **General Responses Q3**. DINO and CLIP-I are computed between the generated images and real images. For image variation, we calculate the CLIP-T between the generated image and the prompt "This is a photo of [Class Name].". For text-edited image variation, we use the editing text as the prompt, such as "A [Class Name] with a mountain in the background.". We report the results of individually fine-tuning the two parts, referred to SD-IPC-FT (C) for the CLIP part and SD-IPC-FT (U) for the U-Net part.
> The qualitative results of image variation are in Rebuttal Figure 2, here is the quantitative results:
>
> | Method | DINO | CLIP-I | CLIP-T |
> | :--- | :---: | :---: | :---: |
> SD-IPC | 44.60 | 77.44 | 25.47 |
> SD-IPC-FT (C) | 49.11 | 76.51 | 25.82 |
> SD-IPC-FT (U) | 48.53 | 79.06 | **26.17** |
> SD-IPC-FT | **52.03** | **79.59** | 25.90 |
>
> The qualitative results of text-edited image variation are Rebuttal Figure 3, the quantitative results are below:
>
> | Method | DINO | CLIP-I | CLIP-T |
> | :--- | :---: | :---: | :---: |
> SD-IPC | 31.09 | 68.66 | 26.84 |
> SD-IPC-FT (C) | 29.10 | 67.03 | 27.99 |
> SD-IPC-FT (U) | 35.21 | 69.99 | 28.56 |
> SD-IPC-FT | **40.28** | **71.97** | **28.69** |
>
> As depicted in the two tables, fine-tuning both the CLIP prompts and U-Net layers contribute to achieving better results.
> ___
> ### Q2: Why not SD-IPC adopts the backbone of SD-R, i.e., Stable Diffusion v2.1?
> We appreciate your suggestion. Initially, we followed Custom Diffusion [1], which employs Stable Diffusion v1.4, as a starting point for developing our method. SD-R [2] was made available in March 2023, subsequent to our method's development. In response to your point, we have ascertained that SD-IPC is compatible with Stable Diffusion v2.1 as well, and the variation results can be found in Rebuttal Figure 5.
> ___
> ### Q3: In Sec.4.3, I suggest the paper adopts the dataset and edited prompt used by DreamBooth to make a full comparison. Both qualitative and quantitative results should be presented.
> Thank you for pointing out this issue, we have addressed this matter in **General Responses Q2**, where we present the experiments of customized generation. The qualitative comparisons are in Rebuttal Figure 1. We will incorporate these results into our final version and provide more visual results in the supplementary material.
> ___
> ### Q4: It would be better if the quantitative results of the first ablation experiment are presented.
> Thank you for the constructive comments. The ablation study is reported in **Q1**.
> ___
> ### Q5: The motivation for applying deep prompts tuning in SD-IPC-FT needs to be further explained.
> Given our limited training data consisting of only 100 images for fine-tuning the Stable Diffusion [1], directly adjusting every parameter could potentially result in severe overfitting and the loss of previously acquired knowledge. In light of this, a solution emerged from prior research [3], where deep prompt-tuning exhibited to be able to resist overfitting. On the other hand, the fine-tuning aims to address inferior CLIP feature, and to change the focus on different aspects of the feature (object, scene, or portrait). Consequently, we have chosen to adopt this approach.
>
> ___
> [1] Kumari, Nupur, et al. "Multi-concept customization of text-to-image diffusion." Proceedings of the IEEE/CVF Conference on Computer Vision and Pattern Recognition. 2023.
>
> [2] Rombach, Robin, et al. "High-resolution image synthesis with latent diffusion models." Proceedings of the IEEE/CVF conference on computer vision and pattern recognition. 2022.
>
> [3] Zhou, Ziqin, et al. "Zegclip: Towards adapting clip for zero-shot semantic segmentation." Proceedings of the IEEE/CVF Conference on Computer Vision and Pattern Recognition. 2023.

---

> > ### Author Response · Authors · 2023-08-17
> >
> > Dear Reviewer hEWJ:
> >
> > Thank you for reviewing our paper. Just a friendly reminder that **the author-reviewer discussion will close soon**, and we eagerly await your feedback. In response to your comment, we've updated the comprehensive ablation studies in (text-edited) image variation tasks. Additionally, we have presented the updated quantitative and qualitative results from the Customized Generation Benchmark. Could you please take a look at these updates?
> >
> > We're here to discuss any more questions or concerns you may have about our paper.
> >
> > With warm regards,
> >
> > Authors

---

> > > ### Comment · Reviewer_hEWJ · 2023-08-20
> > >
> > > After reading other reviewers' comments and the rebuttals, I raise my rating.

---

### Official Review · Reviewer_aD4y · 2023-07-06

**Soundness:** 3 good
**Presentation:** 2 fair
**Contribution:** 2 fair
**Rating:** 5
**Confidence:** 3

**Summary:**

This paper focuses on the problem of generating images by a reference image w/ or w/o further text guidance. The core of this problem partly lies in how to convert the images to embeddings that can be directly feed into Stable Diffusion model. To this end, the authors propose to leverage CLIP model, the text encoder of which is also part of the stable diffusion model. They compare results with Stable Diffusion Re-imagination, as well as Custom diffusion.

**Strengths:**

The strengths are as follows:

- This is a timely interesting topic. How to generate new images by a referring image has been an interesting problem, especially given the recent burst of diffusion models. Compared to previous method such as Dreambooth, this method is light-weight. Authors even provided a version without any fine-tuning, though the performance may not be that good. But the authors also provide a easy-to-finetune pipleline which seems to largely improve the performance

- The training cost is much lower. Compared to SD-R which requires 200,000 GPU-hours, this method here reduced the cost to only 100 images with 1 GPU-hour. Compared to Custom Diffusion which requires 250 iterations of fine-tuning, this paper only required ~30 steps.

- This paper provides the generation capability both w/ or w/o text, in addition to the reference image. This is a nice property.

- The results look reasonable.

**Weaknesses:**

The first thing to improve might be the writing.

(1) The abstract part can be improved. When I first read the abstract without knowing the main text, I was a bit struggling to understand which exactly problem this paper is particularly trying to solve. Converting image to text? Improving Diffusion? or Image-to-Image generation? This part can be clearer.

(2) There is actually not much contents discussing related work, despite the existence of section 2. I would recommend authors have a separate related section, that discuss prior work sufficiently. For example, the current 2.3 section only discussed SD-R and just mentioned dreambooth, textual inversion, and custom diffusion. But there are more works, such as instructpix2pix, plug-and-play, etc. The similarities and differences can be articulated. Besides, much background contents in section 2 cam actually be cut, e.g., line 81-84.

(3) The formulation in Section 3 can be improved. E.g., where is the 76 comes from? This is not obvious for people unfamiliar with the CLIP details. When you combine text prompt and converted image prompt in equation (8), how do you do that clearly?

(4) you can have a figure illustrating the fine-tuning process in section 3.2

The next question I have is that, can this method be generally applied to other text-to-image diffusion models. It seems to me that this paper exploits the prior knowledge that stable diffusion used CLIP text encoder, thus you can use text image encoder to convert image to text prompt. This is fine, but would this still work with other approaches such as imagen that relies on T5 model? e.g. dreambooth is agnostic to the text encoder and thus is more generic.

While the authors claimed this method is cost effective compared to the the SD-R trained on millions of images and over 200,000 GPU hours, has the author measured the out-of-distribution capability sufficiently? e.g. SD-R may just train once, and generalize to all images, but this method needs to be trained for each domain.

Detailed question:

(1) it seems the way you combine text and image prompts in eq (8) is adhoc, what if the text tokens are longer than 76, then where do you put those $f_{txt}^{comb}$? This seems technically problematic.
(2) For line 133, you assumed the norm to be 27 as a constant, which seems also problematic to me. This feels not rigorous. One better way to justify this is to visualize the histograms of the norms.
(3) for imagenet, cele-A, and places fine-tuning using equation (9), how do you get the text, image pair for the $L_{text}$?


Results section:

(1) directly compared the SD-IPC-FT w/ text version with SD-R is apple-to-orange. The SD-R just uses image as guidance, you should compare the SD-IPC ( w/ or w/o FT) version that only uses images as well. You can have w/ and w/o text results both there.
(2) No quantitative comparison between SD-IPC-FT with custom diffusion.
(3) seems SD-R without text input is still better in Table 2?
(4) custom diffusion accept multiple concepts, but seems this method is not able to.
(5) In figure 9, for SD-IPC-FC, what if you initialize the fc layer from the closed form inverse projection, but make it trainable?

**Questions:**

see above

**Limitations:**

I guess two obvious limitation are:
(1) may only work for text-to-image models that relies on CLIP encoder
(2) need to get re-trained every time for a new domain.

---

> ### Author Rebuttal · Authors · 2023-08-10
>
> Thank you for your suggestion. The common questions are first answered in **General Responses**. Below please find the responses to specific comments.
> ___
> ### Q1: The abstract part can be improved.
> We appreciate your suggestion. We will rewrite the abstract to highlight the relationship of the proposed methods, and the relationship with other methods, as summarized in **General Responses Q1**.
> ___
> ### Q2: Separate related section, that discuss prior work sufficiently.
> We will incorporate a discussion in the final version. We have included some quantitative experiments in **General Responses Q2**. It's important to note that, as explained in **General Responses Q1**, our method is primarily focus on image variation (producing similar images to the reference image) and customized image generation. As a result, direct comparisons with image editing techniques such as instructpix2pix, plug-and-play may not be applicable.
> ___
> ### Q3: Questions about the text token length.
> The number 76 derives from the CLIP, where the text's maximum length is 77 (token index is from 0 to 76). The sentence would be padded or truncated to length of 77. This is also followed by the Stable Diffusion [1]. In the context of Eq.(8), the combination is computed by adding the end-token embedding with a weight $\alpha$ to the converted image embedding, as $f_{txt}^{comb} = f_{txt}^{cnvrt} + \alpha \cdot f_{txt}^{t,\left\langle {eos} \right\rangle }$. After the combination, the $f_{txt}^{comb}$ would replace the end-token and all pad-tokens.
> ___
> ### Q4: Figure illustrating the fine-tuning process.
> We appreciate your valuable suggestion. The framework of our method has been presented in the supplementary material Section A. In response to your comment, we will improve the figures to provide clearer insights.
> ___
> ### Q5: Can this method be generally applied to other text-to-image diffusion models?
> Our method is compatible with the diffusion model utilizing CLIP as text encoder. As Stable Diffusion is now the dominating text-to-image generation model, our work has sufficient impact to the related research community.
> ___
> ### Q6: SD-R may just train once, but this method needs to be trained for each domain.
> Our method shows out-of-distribution generalization in experiment results. Notably, the testing images featured in the paper remain entirely separate from the fine-tuned images. Furthermore, the fine-tuning of SD-IPC-FT on object, scene, and portrait serves to **define what to preserve from the reference image**. This is a **distinct advantage of our method** as image variation inherently has some ambiguity in deciding what to preserve from the reference image. In contrast, SD-R lacks this capability, as shown in Paper Figure 4. SD-R struggles to preserve scene information in the "Bar" example (it still focuses on the crowd). In other words, our fine-tuning is **not on individual domains**, but rather to **addressing diverse requirements of preservation**.
> ___
> ### Q7: Visualize the histograms of the norms.
> The histograms depicting the norms of embeddings can be found in Rebuttal Figure 4. Specifically, the norms of a randomly chosen set of 1,000 texts ranges from 26.5 to 29.
> ___
> ### Q8: For imagenet, celeb-A, and places fine-tuning, how do you get the text?
> The texts of ImageNet and Places365 are class name prompt "This is a photo of [Class Name].". For portrait, we utilize the MM-CelebA-HQ dataset which has image captions for each portrait.
> ___
> ### Q9: Directly compared the SD-IPC-FT w/ text version with SD-R is apple-to-orange.
> We have indeed compared both SD-IPC-FT **w/ text** and SD-R **w/ text** in Paper Figure 5 and Figure 6, where the samples labeled as "Without Editing" showcase results only from images, while the other samples encompass edited text, such as "On the beach" and "Under the starry sky". It is evident from figures that our SD-IPC-FT surpasses SD-R in terms of editing performance. For a quantitative comparison, refer to **General Responses Q3**. Apart from the "Wearing a hat" instance of the male in Paper Figure 6, SD-R exhibits minor edits. This circumstance might mislead the reviewer to think them as SD-R results without editing.
> ___
> ### Q10: No quantitative comparison between SD-IPC-FT with custom diffusion.
> Thank you for your suggestion. We have reported the quantitative comparison in **General Responses Q2** and visual results in Rebuttal Figure 1.
> ___
> ### Q11: SD-R without text input is still better in Table 2.
> In Table 2, we aimed to illustrate that the training-free SD-IPC achieves comparable results to SD w/ Text (Line. 216). Notably, excessively high CLIP score may indicate a lack of variation, while too low score suggests fidelity loss in the generated image. The SD w/ Text effectively encapsulates all desired semantics from the reference image. Our method achieves a similar score to SD w/ Text, demonstrating its ability of substantial variation while maintaining sufficient semantics. Our method may exhibit lower image quality than SD-R, due to the backbone difference, V1.4 vs. V2.1, ours still outperforms SD-R in editing performance, as evident in **General Responses Q3**. Also shown in Rebuttal Figure 5, our method works for SD V2.1.
> ___
> ### Q12: For SD-IPC-FC, initialize the fc layer from the closed form inverse projection.
> Following the benchmark in **General Responses Q3**, we evaluate the image variation, where the CLIP-T is between the generated image and a prompt "This is a photo of [Class Name].". SD-IPC-FC (I) is to initialize the FC layer with inverse projection. The initialization can alleviate the overfitting, but only with limited effectiveness. Visual results are in Rebuttal Figure 2, quantitative comparisons are as followed:
> | Method | DINO | CLIP-I | CLIP-T |
> | :--- | :---: | :---: | :---: |
> SD-IPC | 44.60 | 77.44 | 25.47 |
> SD-IPC-FT | **52.03** | **79.59** | **25.90** |
> SD-IPC-FC | 24.71 | 64.96 | 23.76 |
> SD-IPC-FC (I) | 46.55 | 76.54 | 25.78 |
> ___

---

> > ### Author Response · Authors · 2023-08-17
> >
> > Dear Reviewer aD4y:
> >
> > Thank you for reviewing our paper. Just a friendly reminder that **the author-reviewer discussion will close soon**, and we eagerly await your feedback. In response to your comment, we've updated the quantitative and qualitative results of the Customized Generation Benchmark, which highlights our method's advantage in terms of fewer training iterations. We also answered the questions about details in your comments. Besides, the misunderstanding of the apple-to-orange comparison is clarified. Could you please take a look at these updates?
> >
> > We're here to discuss any more questions or concerns you may have about our paper.
> >
> > With warm regards,
> >
> > Authors

---

> > ### Comment · Reviewer_aD4y · 2023-08-20
> > **Thank your for the clarification**
> >
> > Dear Authors,
> >
> > Thank you for the rebuttal and the efforts to clarify my questions. This does address some of my concerns.
> >
> > I think my concern still centers around the Q5 and Q6.
> >
> > > Our method is compatible with the diffusion model utilizing CLIP as text encoder. As Stable Diffusion is now the dominating text-to-image generation model, our work has sufficient impact to the related research community.
> >
> > I think what you said makes sense, and I agree that Stable Diffusion is a major impact at this point. But the concern is also valid, as others are emerging/has emerged, e.g., Deep-Floyd/IF and Imagen, which used T5 embedding.
> >
> > > Notably, the testing images featured in the paper remain entirely separate from the fine-tuned images
> >
> > Are they from different domains or the same domain?
> >
> > > In contrast, SD-R lacks this capability, as shown in Paper Figure 4. SD-R struggles to preserve scene information in the "Bar" example (it still focuses on the crowd).
> >
> > I hold a different opinion about Figure 4. It seems obvious to me that, SD-R is better than SD-IFC (yours). Of course SD-IFC is not fine-tuned, but it changed the bear to a baby, and not as loyal to the reference image as SD-R in terms of pose and angles. In terms of the "Bar" example, I actually felt SD-R is better as it maintains the vibe while SD-IFC switched it to a family gathering feeling.
> >
> > Q: can you release your code and model if the paper is accepted?

---

> > > ### Author Response · Authors · 2023-08-20
> > >
> > > Dear Reviewer aD4y,
> > >
> > > We appreciate your reply, For your concerns.
> > >
> > > 1. We agree with your point. Our method is rooted in the relationship between CLIP and Stable Diffusion. The current solution is not directly applicable to T5-style models. However, some parts of our methods, e.g., the conversion from CLIP image embedding to text embedding, and the importance of a good initialization, might offer inspiration to models or fields beyond the Stable Diffusion.
> > >
> > > 2. The test images belong to the same domain (ImageNet, Places, and CelebA) as the 100 images used for fine-tuning, yet they are from different classes. We also tried to generate CelebA images with ImageNet fine-tuned model and vice versa, it shows that they are able to produce reasonable results. We could report some results in the final supplementary material.
> > >
> > > 3. We want to clarify that Paper Figure 4 DOES NOT intend to argue SD-IPC outperforms SD-R, it signifies the superior performance of SD-IPC-FT compared to SD-R and SD-IPC. SD-IPC only serves as an initial experiment that motivates the development of our better method. We are aware of its limitations, for example, they can fail to discriminate the semantically related concepts, like "kids" and "teddy bear". We have mentioned this in Line 219 in the original paper.
> > >
> > > 4. Yes, we will release the code and demo if the paper is accepted.
> > >
> > > With warm regards, Authors

---

> > > > ### Comment · Reviewer_aD4y · 2023-08-20
> > > >
> > > > Thank you for the clarification. Makes sense.

---

### Official Review · Reviewer_1WWU · 2023-07-06

**Soundness:** 4 excellent
**Presentation:** 4 excellent
**Contribution:** 4 excellent
**Rating:** 7
**Confidence:** 3

**Summary:**

This paper titled presents a method called Stable Diffusion Image-to-Prompt Conversion (SD-IPC) that leverages the inherent capabilities of the Contrastive Language-Image Pre-Training (CLIP) model to convert images into text prompts for image generation tasks. The authors start from the analysis that the control of image generation through text is primarily influenced by the embedding of the end-of-sentence (EOS) token, and masking other word tokens does not significantly affect the quality of image generation. From this constatation, they derive a closed-form projection matrix that converts visual embeddings into text embeddings and use them to control the Stable Diffusion image generation process.

**Strengths:**

 **Technical soundness** The paper is well written and the explanation of the underlying mechanisms of the CLIP model and its relationship with image generation is made intuitive.

**Simple and clever visual-to-prompt embedding conversion** The paper presents a straightforward yet ingenious approach to convert images into text prompts by inverting the textual projection layer. This conversion allows for image variations and editing, effectively bridging the gap between images and textual prompts. The simplicity and cleverness of this method contribute to its practicality and usability.

**Efficient variation model learning through fine-tuning** The authors propose two methods to enhance the quality and flexibility of image-to-prompt conversion. The first method involves parameter-efficient tuning using a small amount of data, requiring minimal computational resources and time. This approach enables efficient learning of the variation model, making it practical for real-world applications.

**Visually sound generations and variations** The paper showcases visually sound image generations and variations achieved through the proposed SD-IPC method. By leveraging the inherent capabilities of the CLIP model, the generated images align well with the desired concepts and demonstrate high-quality results.

**Weaknesses:**

The method is only compared with SD-R and the inclusion of additional comparisons with existing methods could have further strengthened the paper in particular for the evaluation of the editing capability of the method.

**Questions:**

As the authors said, the approximation of the inversion of the textual projection matrix may be a source of suboptimal results. How does this impact the generation process?

**Limitations:**

The authors show some limitations in their work including the need for coherence between the editing prompt and the target image or the lack of multiple target images.

---

> ### Author Rebuttal · Authors · 2023-08-10
>
> Thank you for the positive comments. Some common questions are first addressed in the **General Responses**, followed by answers to individual reviews.
> ___
> ### Q1: Additional comparisons with existing methods could have further strengthened the paper in particular for the evaluation of the editing capability of the method.
> Thanks for your suggestion. The experiments of customized generation have been detailed in **General Responses Q2**. Both quantitative and qualitative results will be incorporated into our final version.
> ___
> ### Q2: As the authors said, the approximation of the inversion of the textual projection matrix may be a source of suboptimal results. How does this impact the generation process?
> We express our gratitude to the reviewer for raising this important concern. The discrepancy between the projected visual feature and its textual counterpart indeed introduces confusion between concepts. As evident in Paper Figure 4 and Line 219$\sim$221, an image of a "teddybear" was transformed into a picture of a "kid", as also demonstrated in Supplementary Material Figure 15, where a "bear" became a "tiger". The primary driver of this confusion stems from the inherent modality gap between images and text. Our CLIP experiments have revealed the cosine similarity between ground-truth image-text pairs is only around 0.3, which signifies the alignment of features is not perfect. This modality gap also persists after the inverse matrix projection, leading to the perplexing shifts between closely related concepts.
>
> Motivated by this gap, we propose SD-IPC-FT approach to reduce the gap via fine-tuning partial parameters of the SD model.
> ___

---

> > ### Comment · Reviewer_1WWU · 2023-08-20
> >
> > Thank you for your detailed answers.
> >
> > I am happy with the paper and will keep my rating.

---

### Official Review · Reviewer_rWok · 2023-07-07

**Soundness:** 3 good
**Presentation:** 3 good
**Contribution:** 2 fair
**Rating:** 5
**Confidence:** 4

**Summary:**

This paper demonstrates that the CLIP model, used in Stable Diffusion, inherently possesses the ability to convert images into text prompts. They achieve this by utilizing a linear projection matrix calculated in a closed form. Furthermore, the paper shows that this capability can be enhanced by incorporating a small amount of similar-domain training data or performing online training steps on reference images. These approaches offer a simple and flexible solution to bridge the gap between images and text prompts, enabling more effective interaction between the two modalities.

**Strengths:**

1. This paper is well organized and the motivation is good. The basic idea of Image-to-Prompt Conversion is simple but interesting. I like the idea of using a closed-form projection matrix to converts the visual embeddings into the semantic prompt which can help control the stable diffusion.
2. The insights about start-/end-token's attention map for stable diffusion are impressive.
3. The proposed methods' finetuning time of customized generation is significantly faster than the current methods.

**Weaknesses:**

1. Although the basic idea of the paper is cool, the visual results of SD-IPC and SD-IPC-FT do not have sufficient preponderance over than previous methods.
2. This paper only compares proposed methods with Customized Generation and SD-R. I think the authors should also add experiments to compare the SD-IPC-FT with DreamBooth (with and w/o LoRA), textual inversion and Reference-only module for controlnet.
3. As shown in Table 2, SD-IPC achieves worst FID which means they output the most unrealistic methods. Besides, the CLIP-Score are also not good enough. I think authors should add more explanations of these tables in the SD-IPC part in section 4.2. Aslo, more comparisons should be added in these tables.

**Questions:**

1. Since SD-IPC and textual inversion all aims to convert the visual embeddings into the text prompt, detailed comparison with it should be included in this part. From my point of view, SD-IPC is just a simplified textual inversion method.
2. For Customized Generation, more comparisons should be provided in this paper.

**Limitations:**

Not applicable.

---

> ### Author Rebuttal · Authors · 2023-08-10
>
> Thank you for the constructive comments. The common questions are first answered in **General Responses**, then we clarify questions from individual review.
> ___
> ### Q1: The visual results of SD-IPC and SD-IPC-FT do not have sufficient preponderance over than previous methods.
> SD-IPC-FT surpasses prior methods like SD-R [1] in terms of editing performance. SD-R [1] occasionally struggles with effective editing, as shown in Paper Figure 5, 6, and Rebuttal Figure 3. Quantitative results in **General Responses Q3** reinforce this point, with our method achieving an editing fidelity (CLIP-T) of 28.69, compared to SD-R [1] at 26.01. Notably, even the training-free SD-IPC attains a CLIP-T score of 26.84.
> ___
> ### Q2: I think the authors should also add experiments to compare the SD-IPC-FT with DreamBooth (with and w/o LoRA), textual inversion and Reference-only module for controlnet.
>
> Thank you for pointing out this issue, we have addressed this matter in **General Responses Q2**, where we present the experiments of customized generation. Our SD-IPC-CT is comprehensively compared with DreamBooth [2], Textual Inversion [3], and Custom Diffusion [4]. The visual results are depicted in Rebuttal Figure 1. We will incorporate these results into our final version and provide more visual results in the supplementary material. We acknowledge your interest regarding a comparison with the recently introduced Reference-only controlnet. However, we must clarify that this falls outside the scope of our paper, as the Reference-only controlnet was released in May 2023, coinciding with the time of our submission.
> ___
> ### Q3: As shown in Table 2, SD-IPC achieves worst FID which means they output the most unrealistic methods. Besides, the CLIP-Score are also not good enough.
> Thank you for the comment, it appears that the reviewer misunderstood the purpose of Table 2: Table 2 is used to demonstrate the effectiveness of the training-free image-to-prompt conversion equation, which severs as the initialization of SD-IPC-FT and SD-IPC-CT.
> In Table 2, we can see that SD-IPC achieves comparable results to SD with ground-truth text (Line. 216). The CLIP score, which compares generated and reference images, quantifies their "content distance". It's important for the CLIP score to fall within a reasonable range, excessively high scores may indicate a lack of variation, (For example, if the generated image is identical to the to-be-varied image (no variation effect), it will result in the highest CLIP score) while too low scores suggest fidelity loss in the generated image. When referring to SD with ground-truth text prompt (SD w/ Text), it means generating images using the text-prompt of reference image. Consequently, the generated result effectively encapsulates all desired semantics from the reference image. Our method achieving a similar CLIP score to SD w/ Text demonstrates its ability to generate substantial variation while maintaining sufficient semantics. While our method may exhibit lower image quality than SD-R (partly due to the backbone difference, V1.4 vs. V2.1), our approach still outperforms SD-R in editing performance, as evident in **General Responses Q3**. As shown in Rebuttal Figure 5, our method also works for Stable Diffusion V2.1, we will update our results in the new version.
> ___
> ### Q4: From my point of view, SD-IPC is just a simplified textual inversion method.
> As stated in **General Responses Q1**, SD-IPC and SD-IPC-FT are designed for image variation generation, focusing on generating similar semantics as the reference, e.g., similar scene or object. Textual Inversion [3] aims customized image generation, preserving the reference object's identity. Notably, our SD-IPC-CT also enables fast customized image generation. Furthermore, **General Responses Q2** expands more comparisons beyond the results shown in Figure 7 of the current paper.
> ___
> ### Q5: For Customized Generation, more comparisons should be provided in this paper.
> We appreciate your suggestion. The experiments related to customized generation have been detailed in **General Responses Q2**. Both quantitative and qualitative results will be incorporated into our final version.
> ___
> [1] Rombach, Robin, et al. "High-resolution image synthesis with latent diffusion models." Proceedings of the IEEE/CVF conference on computer vision and pattern recognition. 2022.
>
> [2] Ruiz, Nataniel, et al. "Dreambooth: Fine tuning text-to-image diffusion models for subject-driven generation." Proceedings of the IEEE/CVF Conference on Computer Vision and Pattern Recognition. 2023.
>
> [3] Gal, Rinon, et al. "An Image is Worth One Word: Personalizing Text-to-Image Generation using Textual Inversion." The Eleventh International Conference on Learning Representations. 2022.
>
> [4] Kumari, Nupur, et al. "Multi-concept customization of text-to-image diffusion." Proceedings of the IEEE/CVF Conference on Computer Vision and Pattern Recognition. 2023.

---

> > ### Author Response · Authors · 2023-08-17
> >
> > Dear Reviewer rWok:
> >
> > Thank you for reviewing our paper. Just a friendly reminder that **the author-reviewer discussion will close soon**, and we eagerly await your feedback. In response to your comment, we've updated the quantitative and qualitative results of the Customized Generation Benchmark, which highlights our method's advantage in terms of fewer training iterations. Besides, we also explained how it's different from the Textual Inversion. Could you please take a look at these updates?
> >
> > We're here to discuss any more questions or concerns you may have about our paper.
> >
> > With warm regards,
> >
> > Authors

---

> > > ### Comment · Reviewer_rWok · 2023-08-20
> > > **Official Comment**
> > >
> > > Thanks authors for the detailed rebuttal. It solves most of my concerns. Thus, I will raise my rating.

---

### Author Rebuttal · Authors · 2023-08-10

# General Responses
We thank all reviewers for your thoughtful and detailed feedback, which is of great importance to our work. Here we address some common issues and share some findings that all reviewers might have interests in.
___
### Q1: Clarification of certain facets of the proposed method.
This paper presents a novel finding that allows for the direct transformation of an image into a textual prompt embedding in the CLIP model [1]. Leveraging this finding, we formulate a range of methods with distinct advantages and applications. In the subsequent sections, we highlight the certain facets of our methods:

 Method | Application | Comparisons | Merits | Relationship |
 :--- | :--- | :--- | :--- | :--- |
 SD-IPC | Image Variation | SD-R [2] | Training-free, Demonstrate the finding | Serving as the initiation of SD-IPC-FT and SD-IPC-CT |
 SD-IPC-FT | Image Variation | SD-R [2] | Lightweight training, Able to customize what to preserve from reference images | Tuning SD-IPC offline, Better edit effect than SD-R [2] |
 SD-IPC-CT | Customized Generation | DreamBooth [3], Textual Inversion [4], Custom Diffusion [5] | Fast online training, Better generation quality with few training iterations | Tuning SD-IPC online |
___
### Q2: Results of customized generation benchmark and the comparison with common methods.
Following the benchmark proposed in DreamBooth [3], we employ a dataset comprising 30 subjects belonging to 15 different classes for the fine-tuning. Each subject is represented by 4$\sim$6 images, and will be edited using a total of 25 texts for 4 times, getting a results of 3,000 images. DINO and CLIP-I score are computed for subject fidelity (preservation of subject identity details), CLIP-T score is for prompt fidelity (the editing performance). We compare our SD-IPC-CT with three common customized generation methods: DreamBooth [3], Textual Inversion [4], and Custom Diffusion [5]. The visual results are depicted in Figure 1 of Rebuttal PDF, the quantitative results obtained are as followed:

| Method | DINO | CLIP-I | CLIP-T | Comments |
| :--- | :---: | :---: | :---: | :--- |
| DreamBooth [3] | **60.11** | **77.78** | 25.81 | **Good Identity**, Weak Editing |
| Textual Inversion [4] | 25.11 | 62.44 | 29.53 | Weak Identity, **Good Editing** |
| Custom Diffusion [5] | 39.67 | 68.37 | **30.90** | Weak Identity, **Good Editing** |
| SD-IPC-CT (Ours) | 50.25 | 74.59 | 28.14 | **Good Identity**, **Good Editing** |

The result demonstrates that DreamBooth [3] obtains the highest DINO/CLIP-I scores. However, its CLIP-T score is significantly lower, indicating unsatisfactory editing performance. This is also apparent in the visual results, where DreamBooth [3] generated images closely resemble the training images with minor edits. In contrast, Textual Inversion [4] and Custom Diffusion [5] exhibit strong CLIP-T scores, albeit much lower DINO/CLIP-I scores, highlighting their weakness in preserving subject details. In comparison, our SD-IPC-CT method strikes a good balance between subject identity preservation and editing performance.

___
### Q3: The advantage of the proposed method over SD-R [2].
In the current version of paper, we highlight SD-IPC-FT's advantage over SD-R [2] in Paper Figure 5, Figure 6, demonstrating the superior editing performance of SD-IPC-FT. Here we add a quantitative experiment to valid this claim. Specifically, we fine-tune SD-IPC-FT using 100 ImageNet [6] images, ensuring it retains object-level details from a reference image. For testing, we utilize the total of 158 images from DreamBooth benchmark, featuring 30 subjects. To conduct the text-edited image variation task, we randomly select an editing text from DreamBooth benchmark for each test image. Editing performance is evaluated using CLIP-T score. Importantly, DINO and CLIP-I scores are omitted as evaluation since they indicate similarity, which does not reflect the quality of "variation''. For example, if the generated image is identical to the to-be-varied image (no variation effect), it will result in the highest DINO and CLIP-I score. The visual results are shown in Rebuttal Figure 3 (the SD-IPC-FT (C) for only fine-tuning the CLIP part and SD-IPC-FT (U) for only fine-tuning the U-Net part, this two are for ablation study). The quantitative results are as followed:

| Method | CLIP-T |
| :--- | :---: |
SD-IPC | 26.84 |
SD-IPC-FT | **28.69** |
SD-R [2] | 26.01 |

As seen, our method achieving a higher CLIP-T score than SD-R. Furthermore, we include the training-free SD-IPC for comparison, revealing even SD-IPC slightly outperforming SD-R in CLIP-T score.
___
[1] Radford, Alec, et al. "Learning transferable visual models from natural language supervision." International conference on machine learning. PMLR, 2021.

[2] Rombach, Robin, et al. "High-resolution image synthesis with latent diffusion models." Proceedings of the IEEE/CVF conference on computer vision and pattern recognition. 2022.

[3] Ruiz, Nataniel, et al. "Dreambooth: Fine tuning text-to-image diffusion models for subject-driven generation." Proceedings of the IEEE/CVF Conference on Computer Vision and Pattern Recognition. 2023.

[4] Gal, Rinon, et al. "An Image is Worth One Word: Personalizing Text-to-Image Generation using Textual Inversion." The Eleventh International Conference on Learning Representations. 2022.

[5] Kumari, Nupur, et al. "Multi-concept customization of text-to-image diffusion." Proceedings of the IEEE/CVF Conference on Computer Vision and Pattern Recognition. 2023.

[6] Deng, Jia, et al. "Imagenet: A large-scale hierarchical image database." 2009 IEEE conference on computer vision and pattern recognition. Ieee, 2009.

---

### Comment · Area_Chair_U7pH · 2023-08-15
**discussion**

Hi

Thanks again for your service to NeurIPS'23.

The authors have submitted the rebuttal. Could you please look at it and see if your concerns are addressed?

Thanks
Your AC

---

> ### Comment · Area_Chair_U7pH · 2023-08-18
>
> We are already close to the end of the author-reviewer discussion phase. Please engage in the discussion and let us know if your concerns are addressed or if you have any additional concerns.
>
> Thanks,
>
> Regards, AC

---

### Decision · Program_Chairs · 2023-09-21

**Decision:**

Accept (poster)

**Comment:**

This paper studies how to generate images by a reference image with Stable Diffusion. It proposes to leverage CLIP model to convert images into text prompts by utilizing a linear projection matrix calculated in a closed form. The reviewers in general found the proposed method as technical sound, well motivated, interesting and efficient. The initial concerns were mainly about missing comparisons with existing methods and confusion caused by presentation, which were well addressed in the rebuttal. As a result, all reviewers unanimously agree to accept this submission. In the final version, the authors should include all the clarifications and the additional empirical results provided in the rebuttal.